# Fast, high-throughput production of improved rabies viral vectors for specific, efficient and versatile transsynaptic retrograde labeling

Anton Sumser[1], Maximilian Joesch[1], Peter Jonas[1], Yoav Ben-Simon[1,2,3]*

[1]Institute of Science and Technology Austria (ISTA), Klosterneuburg, Austria; [2]Department of Neurophysiology and Neuropharmacology, Vienna Medical University, Vienna, Austria; [3]Allen Institute for Brain Science, Seattle, WA, United States

**Abstract** To understand the function of neuronal circuits, it is crucial to disentangle the connectivity patterns within the network. However, most tools currently used to explore connectivity have low throughput, low selectivity, or limited accessibility. Here, we report the development of an improved packaging system for the production of the highly neurotropic $RVdG_{envA}$-CVS-N2c rabies viral vectors, yielding titers orders of magnitude higher with no background contamination, at a fraction of the production time, while preserving the efficiency of transsynaptic labeling. Along with the production pipeline, we developed suites of 'starter' AAV and bicistronic RVdG-CVS-N2c vectors, enabling retrograde labeling from a wide range of neuronal populations, tailored for diverse experimental requirements. We demonstrate the power and flexibility of the new system by uncovering hidden local and distal inhibitory connections in the mouse hippocampal formation and by imaging the functional properties of a cortical microcircuit across weeks. Our novel production pipeline provides a convenient approach to generate new rabies vectors, while our toolkit flexibly and efficiently expands the current capacity to label, manipulate and image the neuronal activity of interconnected neuronal circuits in vitro and in vivo.

**\*For correspondence:**
yoav.ben-simon@alleninstitute.org

**Competing interest:** The authors declare that no competing interests exist.

## Editor's evaluation

Rabies-mediated monosynaptic retrograde tracing is a powerful method to characterize the connectivity of neural circuits. The CVS-N2c strain of rabies virus shows significantly higher efficiency of transsynaptic spread and less toxicity than the more commonly used SAD B19 strain but has been limited in use by an arduous and lengthy packaging process and low resultant titers. Here, Sumser et al. present a method that significantly speeds up the production process while reducing off-target expression. They also introduce a suite of novel reagents (34 novel plasmids) for monosynaptic tracing with the CVS-N2c strain that they commendably, have already deposited with Addgene. The work is an important advance that will reinvigorate rabies-mediated circuit tracing.

## Introduction

Addressing the complexity and underlying structure of neuronal networks remains one of the biggest challenges in modern neuroscience, as this knowledge is essential for the understanding of circuit functionality in health and disease (*Fornito et al., 2013*; *Morgan and Lichtman, 2013*). Short- and long-range connectivity between populations of neurons in the central and peripheral nervous

systems can be mapped with numerous existing techniques, with varying degrees of simplicity, efficiency and reliability (*Luo et al., 2008*; *Luo et al., 2018*). While recent advances in viral technology, such as anterograde and retrograde AAVs, now enable genetic targeting of populations based on their projection pattern (*Tervo et al., 2016*; *Zingg et al., 2017*), they can only be used for analysis of macrocircuits, due to their lack of specificity. In contrast, monosynaptic tracing technologies, based on engineered rabies virus, enable cell-type specific labeling of presynaptic partners, making it one of the most powerful toolkits to genetically dissect neuronal populations. This process is achieved in two steps: in the first, the avian TVA receptor and the rabies glycoprotein (G) are co-expressed in a predesignated population of cells, usually via administration of an AAV vector (*Figure 1A*) and in the second step, G-deleted rabies viral vectors, pseudotyped with the Avian Sarcoma and Leukosis Virus's envelope glycoprotein A (envA) are introduced to the region containing TVA- and G-expressing neurons (*Figure 1B*), resulting in propagation of rabies particles exclusively from these starter cells to their presynaptic partners, but not to disynaptically connected neurons, and rarely to post-synaptic targets (*Zampieri et al., 2014*), regardless of their physical proximity (*Ginger et al., 2013*; *Wickersham et al., 2007a*; *Figure 1C*).

While this approach ostensibly identifies presynaptic partners associated with a predesignated starter population, inefficiencies in the widely used SAD B19 strain, along with potential transmission biases across cell types (*Albisetti et al., 2017*) suggest the traced cells might represent only a fraction of the complete presynaptic population, rendering smaller, more distributed projections more difficult to identify. Furthermore, since large quantities of native-coat particles are routinely used to initiate the pseudotyping step of the viral production protocol (*Osakada and Callaway, 2013*), background contamination of native-coat particles in the pseudotyped stock is nearly unavoidable, which can result in false identification of projecting populations directly labeled by native-coat particles and not transsynaptically through the starter cells.

Recently, the deployment of the highly neurotropic CVS-N2c rabies strain for monosynaptic tracing was shown to identify 5–20 fold more presynaptic cells per starter cell than SAD B19, enabling a more comprehensive analysis of the diversity of presynaptic cells innervating a predesignated starter population. In addition, this strain was shown to be less neurotoxic, making it more suitable for experiments in behaving animals. However, the production process for these vectors is time-consuming and yields low viral titers (*Reardon et al., 2016*), presumably due to the use of neuronal cell lines for the various amplification and packaging steps. This limitation has so far restricted more widespread use of the superior CVS-N2c vector in circuit mapping experiments, particularly in its more useful pseudotyped form, whose preparation is even lengthier and the titers lower still.

Here, we report the development of a new packaging system, allowing expedited production of high-titer RVdG$_{envA}$-CVS-N2c particles, free of background contamination from native-coat particles, and show that these vectors retain their superior expansion efficacy when compared to SAD B19 vectors. We also report an extended toolkit of AAV and CVS-N2c vectors which can be applied to a wide range of experimental paradigms, and demonstrate their efficacy in uncovering hidden neuronal connections, even in heavily explored circuits.

## Results
### Improved packaging for RVdG viral vectors

Prolonged and cumbersome production has been a major limitation for G-deleted rabies virus vectors and most profoundly so for the CVS-N2c strain. Because CVS-N2c vectors exhibit enhanced neurotropism and retrograde labeling over SAD B19, improvements in the speed and quality of the various N2c production stages are necessary to realize the full potential of rabies viral vectors for mapping synaptic circuits. Here, we introduce a production protocol, based on two new packaging cell lines we have developed: (1) "HEK293-GT" cells, based on HEK293T cells stably expressing the chimeric SAD B19 codon optimized glycoprotein (oG) comprised of the extracellular domain of the Pasteur strain G and the transmembrane domain of the SAD B19G, along with the optimized T7 RNA polymerase (oT7pol), used for the initial rescue and amplification of native-coat particles, and (2) 'BHK-eT' cells, based on BHK21 cells stably expressing the envA glycoprotein, along with the TVA receptor, used for simultaneous vector pseudotyping and amplification (*Figure 1D and E*). The oG was selected for use in the rescue cell line because it can be used to efficiently rescue both the SAD B19 and the CVS-N2c

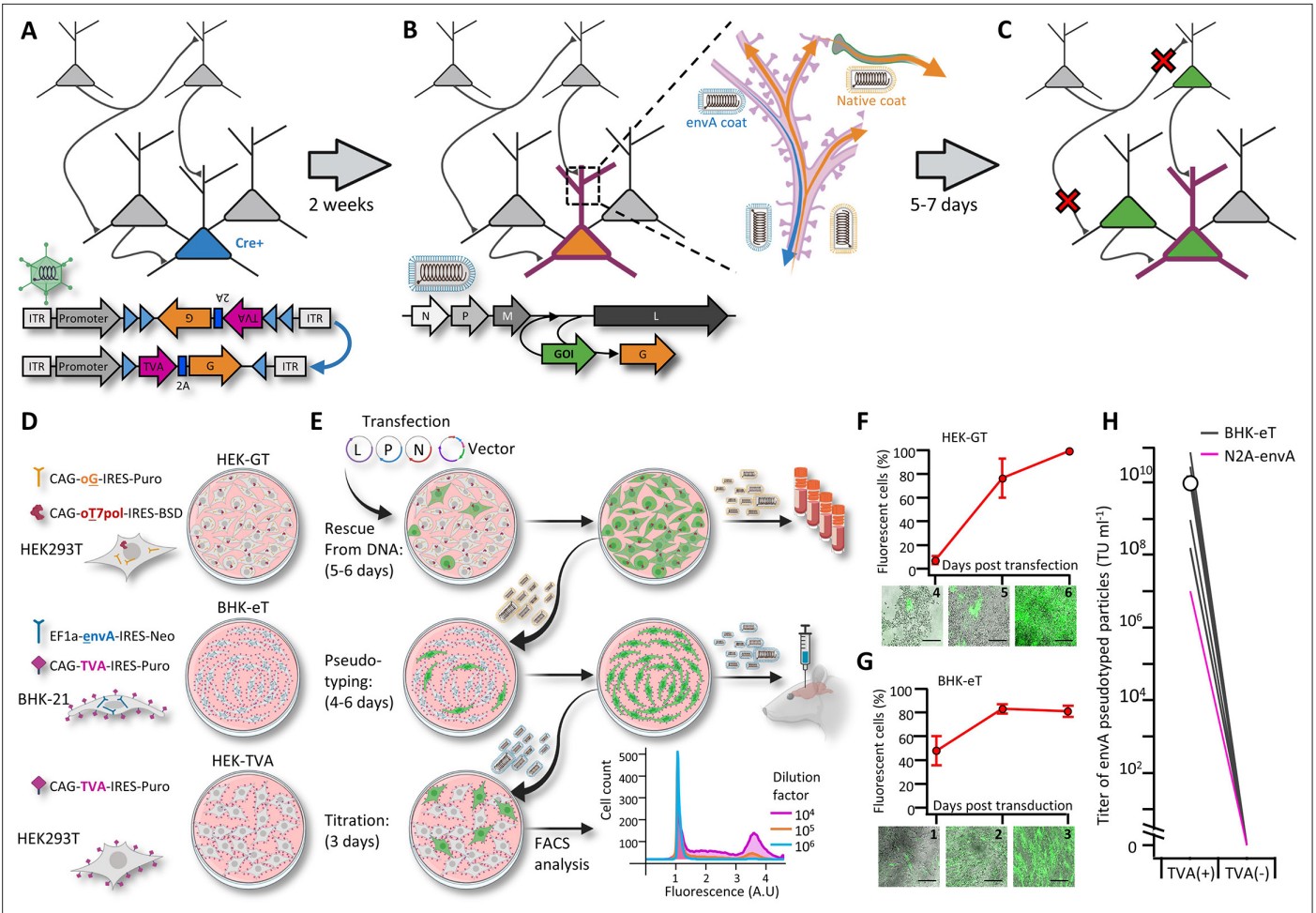

**Figure 1.** A new viral packaging system for fast, clean and high-throughput production of RVdG-CVS-N2c vectors. (**A–C**) A schematic representation of the experimental workflow for achieving cell type-specific trans-synaptic retrograde labeling using G-deleted, envA-pseudotyped rabies viral vectors: Genetic dissection of cre[+] neurons (blue) for conditional expression of TVA and G (**A**); targeting of RVdG$_{envA}$ particles to labeled neurons for expression of a gene of interest (GOI), (**B**) and subsequent propagation of native-coat RVdG particles from starter cells to their presynaptic partners (**C**). (**D**) A schematic representation of the three different cell lines designed for rescue (HEK-GT), pseudotyping and amplification (BHK-eT) and titration (HEK-TVA) of RVdG viral particles, alongside the transgenes used to generate them. (**E**) Schematic representation of the production process and timeline. L,P and N represent the plasmids encoding the corresponding rabies genes and V represents the vector plasmid. (**F**) Quantification of the time course for the rescue stage, starting at day 4 after transfection of viral plasmids. (**G**) Quantification of the time course for amplification of pseudotyped particles, starting at day 1 after transduction with native-coat particles. (**H**) Quantification of the average titer of concentrated pseudotyped stock from 19 individual productions, with comparison to the titer of a representative production of RVdg-CVS-N2c virus, produced using N2a cells (magenta). Lines represent titers of individual productions. Data in F and G represents the average and SEM of three individual and independent measurements.

The online version of this article includes the following source data and figure supplement(s) for figure 1:

**Source data 1.** Quantification of RVdG propagation rates and viral titers.

**Figure supplement 1.** Efficiency of stable cell selection using antibiotic resistance genes.

**Figure supplement 2.** Leak expression of envA-pseudotyped particles can occur around damaged tissue.

**Figure supplement 3.** Effective retrograde labeling with oG coated CVS-N2c particles.

strains and effectively transduce all cell types, whereas particles coated with the N2cG are highly specific to neuronal cell types and do not effectively transduce HEK293 or BHK-21 cells.

To improve the selection process for the transgene carrying cells, both cell lines make use of antibiotic resistance genes and strong constitutive promoters (*Norrman et al., 2018*). This ensures that cultures consist of purely transgene-expressing cells and that expression levels in those cells are high (*Figure 1—figure supplement 1A-C*). Furthermore, unlike previous packaging systems, the co-expression of TVA and envA in the BHK-eT cell line allows pseudotyped particles to propagate within the

**Table 1.** Rescue from DNA and amplification of native-coat stock.

| | B7GG (Osakada and Callaway, 2013) | Neuro2a-N2cG (Reardon et al., 2016) | HEK-GT |
|---|---|---|---|
| Stably-expressed transgenes | T7 polymerase +SAD-B19G | CVS-N2cG | Optimized T7 polymerase (oT7)+optimized SAD-B19G (oG) |
| Selection markers | Fluorescence | Fluorescence | Antibiotic resistance genes |
| Transfected genes | Vector +N,P, G & L | Vector +T7, N,P,G & L | Vector +N,P & L |
| Transfection efficiency | Low | Low | High |
| Growth conditions | 3% CO2 at 35 °C | 3% CO2 at 35 °C | 5% CO2 at 37 °C |
| Rescue timeline | 10–11 days | 10–11 days | 5–6 days |
| Initial amplification timeline | 9–11 days | 14–21 days | Not required |
| Compatibility | SAD-B19 (CVS-N2c) possible, but not tested | CVS-N2c only | Both SAD-B19 and CVS-N2c |

culture, similar to the native-coat ones (*Figure 1F and G*), enabling amplification and pseudotyping of vectors in a single short step. In addition, since a very small amount of native-coat, or even trace amounts of pseudotyped virus, are sufficient in order to initiate the pseudotyping process, contamination from native-coat particles in the final pseudotyped stock can be minimized, while the titer of the pseudotyped stock can be exceedingly high (*Figure 1H*). For each new viral batch produced (37 in total), we injected pseudotyped particles at work concentrations (1–5×$10^8$ TU ml$^{-1}$) into naïve brains. We found only minimal non-specific labeling (a maximum of 1–2 labelled cells per 0.5–1 mm of tissue examined).

Even though the presence of native coat particles can be virtually eliminated from the preparation, it is still possible to see sparse non-specific labeling of pseudotyped particles in the absence of TVA, when extremely high vector titers (100–500-fold higher to our experimental requirements) are introduced into the brain (*Figure 1—figure supplement 2A and B*). To demonstrate that even this sparse labeling likely results from direct penetration of pseudotyped particles into damaged cells and cell processes along the needle tract, rather than from non-specific labeling with native coat particles, we transduced organotypic hippocampal cultures prepared from a WT mouse brain with envA-pseudotyped CVS-N2c-tdTomato, either immediately or one hour after lacerating the culture with a scalpel, to simulate physical damage to neuronal processes (*Figure 1—figure supplement 2C*). We observed that cultures transduced immediately after the insult exhibited substantially higher transduction rates than cultures transduced one hour later, after most damaged processes had time to recover (*Figure 1—figure supplement 2D and E*) confirming that physical damage (such as one that accompanies intracerebral injection with a Hamilton needle), can produce off-target labeling around the injection site, when high-titer virus is delivered. Furthermore, beyond their use as an intermediate

**Table 2.** Pseudotyping of rescued vectors.

| | BHK-EnvA (Osakada and Callaway, 2013) | Neuro2a-envA (Reardon et al., 2016) | BHK-eT |
|---|---|---|---|
| Stably-expressed transgenes | envA or envB | envA | envA +TVA |
| Selection markers | Fluorescence | Fluorescence | Antibiotic resistance genes |
| Growth conditions | 3% CO2 at 35 °C | 3% CO2 at 35 °C | 5% CO2 at 37 °C |
| Pseudotyping timeline | 7–10 days | 28 days | 4–6 days |
| Requirements for pseudotyping | Large stock of native-coat particles | Large stock of native-coat particles | Trace amounts of either native-coat or evA pseudotyped stock |
| Titer | Low 10^8 typical | Low 10^7 typical | High 10^9 typical |
| Native-coat background | 10^2 typical | Not detectable | Not detectable |

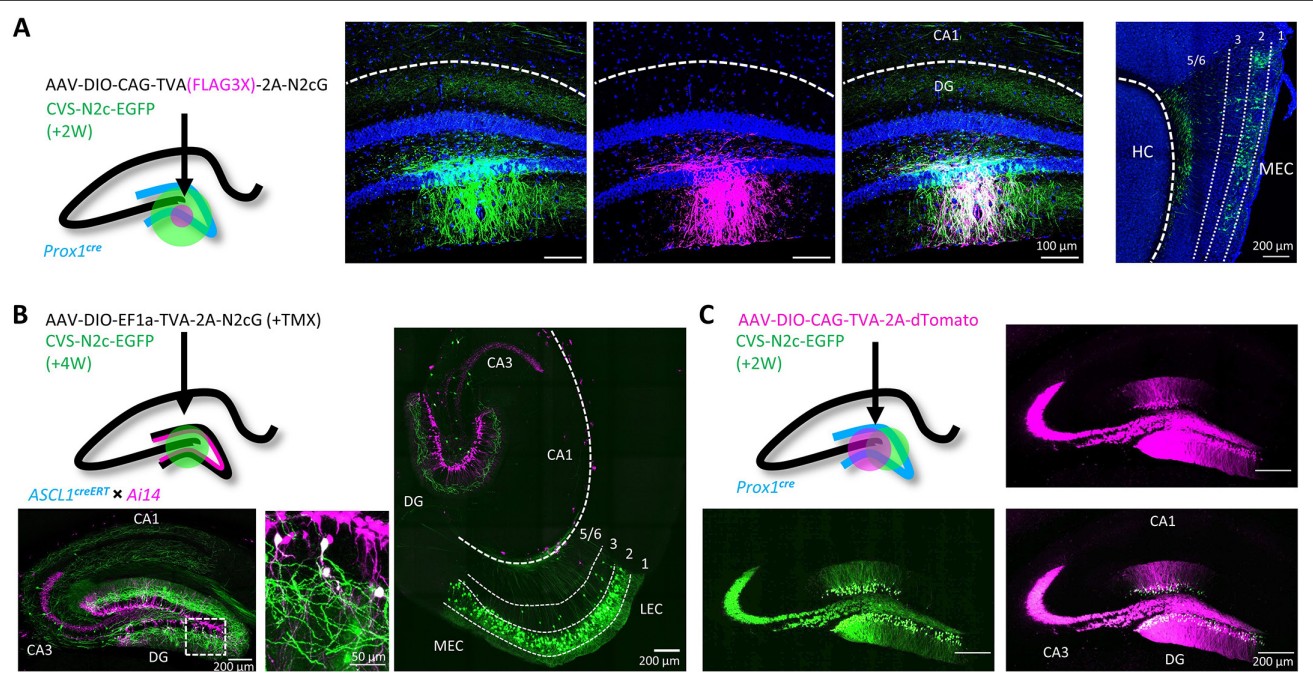

**Figure 2.** Target specificity and retrograde labeling efficacy of envA-pseudotyped RVdG-CVS-N2c particles, produced in BHK-eT cells. (**A**) Illustration of the injection scheme (left) and corresponding representative confocal images demonstrating the targeting specificity of CVS-N2c particles (green) to FLAG3X tagged cells (magenta) along with retrograde labeling to EC layer 2 cells (rightmost image, EC layers are separated by dashed lines and their numbers are denoted above). (**B**) Illustration of the injection scheme (top left) and corresponding representative confocal images demonstrating efficiency and specificity of CVS-N2c-mediated retrograde labeling from adult-born DGCs. (**C**) Illustration of the injection scheme (top left) and corresponding representative confocal images demonstrating target specificity, with no accompanying retrograde labeling, in the absence of the rabies glycoprotein in any of its projections in four separate experiments.

The online version of this article includes the following figure supplement(s) for figure 2:

**Figure supplement 1.** Transsynaptic retrograde labeling pattern from adult-born hippocampal DGCs in subcortical regions.

to obtain envA-pseudotyped particles, we confirmed that oG coated CVS-N2c particles produced in HEK293-GT cells can be successfully used for non-specific retrograde labeling (*Figure 1—figure supplement 3A and B*). Together, these new cell lines enable significantly faster production times, higher titers and less background contamination than both other currently used methods for production of either CVS-N2c or SAD B19 (*Tables 1 and 2*).

## Selective targeting of starter populations

To test the efficacy and transduction exclusivity of these new vectors, we first delivered a minimal volume (20 nl) of AAV vectors expressing a cre-dependent TVA(FLAG3X)–2A-N2cG cassette into the dentate gyrus (DG) of a *Prox1$^{cre}$* mouse (*Borges-Merjane et al., 2020*) followed by large volume (500 nl) of RVdG$_{envA}$-CVS-N2c-EGFP vectors into the same location. Subsequent immunolabeling revealed that only dentate granule cells (DGCs) immunoreactive for the FLAG3X tag were transduced by the rabies vectors, with efficient retrograde labeling from this small group to neurons in the entorhinal cortex (EC, *Figure 2A*; *Borges-Merjane et al., 2020*). In parallel, we performed a second set of experiments in which a TVA-2A-N2cG cassette was expressed in a small population of adult-born DGCs, using an *Ascl1$^{creERT2}$* line crossed with the Ai14 tdTomato cre-reporter line (*Yang et al., 2015*). Four weeks after AAV vectors were delivered to the DG, along with a single i.p. injection of tamoxifen (TMX) to induce recombination in the CreERT2 line, RVdG$_{envA}$-CVS-N2c-EGFP vectors were introduced to the same location. This manipulation resulted in both highly specific targeting of tdTomato$^+$ adult-born DGCs in the inner granule cell layer, as well as robust retrograde labeling of afferents to these starter cells. This high efficacy manifested as widespread and specific labeling of layer-2 neurons of the medial and lateral entorhinal cortices (MEC and LEC, respectively), hilar mossy cells, CA3 pyramidal cells and a number of DG-projecting subcortical regions, including the medial septal (MS), the

supramamilary (SuM), and the raphe (RN) nuclei (*Figure 2B* and *Figure 2—figure supplement 1A* and B), largely consistent with several previous reports (*Deshpande et al., 2013*; *Vivar et al., 2012*). Last, to ensure that the envA pseudotyped virus produced using our system lacks non-specific targeting properties and cannot propagate in the absence of its glycoprotein, we expressed a cre-dependent TVA-2A-tdTomato cassette, without the rabies glycoprotein, in DGCs, again using *Prox1^cre* mice. Subsequent delivery of high-volume (0.5 µl), high-titer (~3 × 10^10 TU ml^-1) RVdG$_{envA}$-CVS-N2c-EGFP vectors resulted in exclusive expression of EGFP in dTomato-positive neurons in the DG (*Figure 2C*) but not in dTomato-negative regions, or in any of the regions projecting to the DG. These experiments provide evidence that the new packaging system we designed is capable of rapidly producing high-titer and high-quality RVdG$_{envA}$ vectors for transsynaptic retrograde neuronal labeling.

## Comparative retrograde labeling efficiency of RVdG strains

RVdG-CVS-N2c vectors were originally produced in the neural progenitor cell line N2A, under the premise that this would improve neurotropism of the assembled viruses, enhancing their trans-synaptic transfer rates and reducing their neurotoxicity (*Reardon et al., 2016*). The new packaging cell lines we developed were designed to increase production speed and efficiency, but the possibility remains that the non-neuronal origin of the BHK-21 cells might compromise the superior properties of CVS-N2c vectors. To evaluate any differences in retrograde labeling efficiency between the vectors assembled using the two approaches, we expressed a TVA-2A-N2cG cassette in CA1 pyramidal neurons and subsequently transduced them with a cocktail of two different RVdG$_{envA}$-CVS-N2c vectors expressing either EGFP or dTomato, in equal titers, amplified and pseudotyped using either N2A-based or the new HEK-GT/BHK-eT based packaging systems (*Figure 3A*). This strategy resulted in highly efficient and specific retrograde labeling of CA1 afferents in the CA3, LEC and medial septum (MS) in equal proportions for both vectors (*Figure 3B, C and H*).

To confirm that these CVS-N2c vectors remain more efficient than the widely used SAD B19 strain, as was previously shown for vectors produced in N2A cells (*Reardon et al., 2016*; *Rowland et al., 2013*), the previous experimental design was repeated, using the optimized B19 glycoprotein (oG; *Kim et al., 2016*) and RVdG$_{envA}$-SAD B19 vectors (*Figure 3D*). Here, a strikingly different result was observed, with afferents labeled with CVS-N2c visibly outnumbering those labeled with SAD B19 in most regions tested (*Figure 3E, F and H*). Quantification of cell numbers in these experiments showed that while the ratio of 1^st order CA1 neurons between the two compared vectors remained low in both experiments (*Figure 3G*), the ratios of 2^nd order neurons in the tested sets of regions differed substantially, with only minor differences observed when the two CVS-N2c vectors were compared, but differences close to an order of magnitude observed when CVS-N2c and SAD B19 vectors were compared (Ipsilateral CA3: 14.43 ± 1.67%; Contralateral CA3: 11.95 ± 2.85%; LEC: 8.17 ± 0.85%; MS: 100.88 ± 4.54%; *Figure 3H*). A notable exception is seen in the projection from the MS, in which ratios between the SAD B19 and N2c remained identical. This effect could be the result of differences in structure of the synaptic contacts between MS and CA1 neurons or in the projection's connectivity scheme. Since we observed that CVS-N2c vectors propagate less efficiently when using the oG, as opposed to their endogenous N2cG (*Figure 3—figure supplement 1A-C*), the actual differences in retrograde labeling between CVS-N2c and SAD B19 are likely to be substantially higher, matching those previously reported (*Reardon et al., 2016*).

## Identification of intra-hippocampal projections to DGCs

To further test and validate the throughput and sensitivity of this tool, we performed additional retrograde labeling experiments from the DG, in order to see whether we will be able to corroborate and expand on previous reports, describing non-canonical inhibitory projections it receives from inhibitory neurons in the *Stratum Oriens* (*S.O.*) and *Stratum Lacunosum Moleculare* (*S.LM.*) of the CA1 (*Hájos and Mody, 1997*; *Katona et al., 2017*; *Klausberger and Somogyi, 2008*; *Szabo et al., 2017*). Those findings are all based on reconstruction of the axonal plexus of biocytin-labeled neurons and, while informative, this approach has low-throughput and mostly lacking information about the identity of the postsynaptic partner. To test if we could locate and identify these cells, we targeted RVdG$_{envA}$-CVS-N2c-tdTomato vectors to DGCs of *Prox1^cre* mice crossed with GAD1-EGFP transgenic mice, in which GABAergic neurons are fluorescently labeled (*Tamamaki et al., 2003*), and examined the regions outside of the DG for colocalization (*Figure 4A*). Consistent with these reports, we found extensive

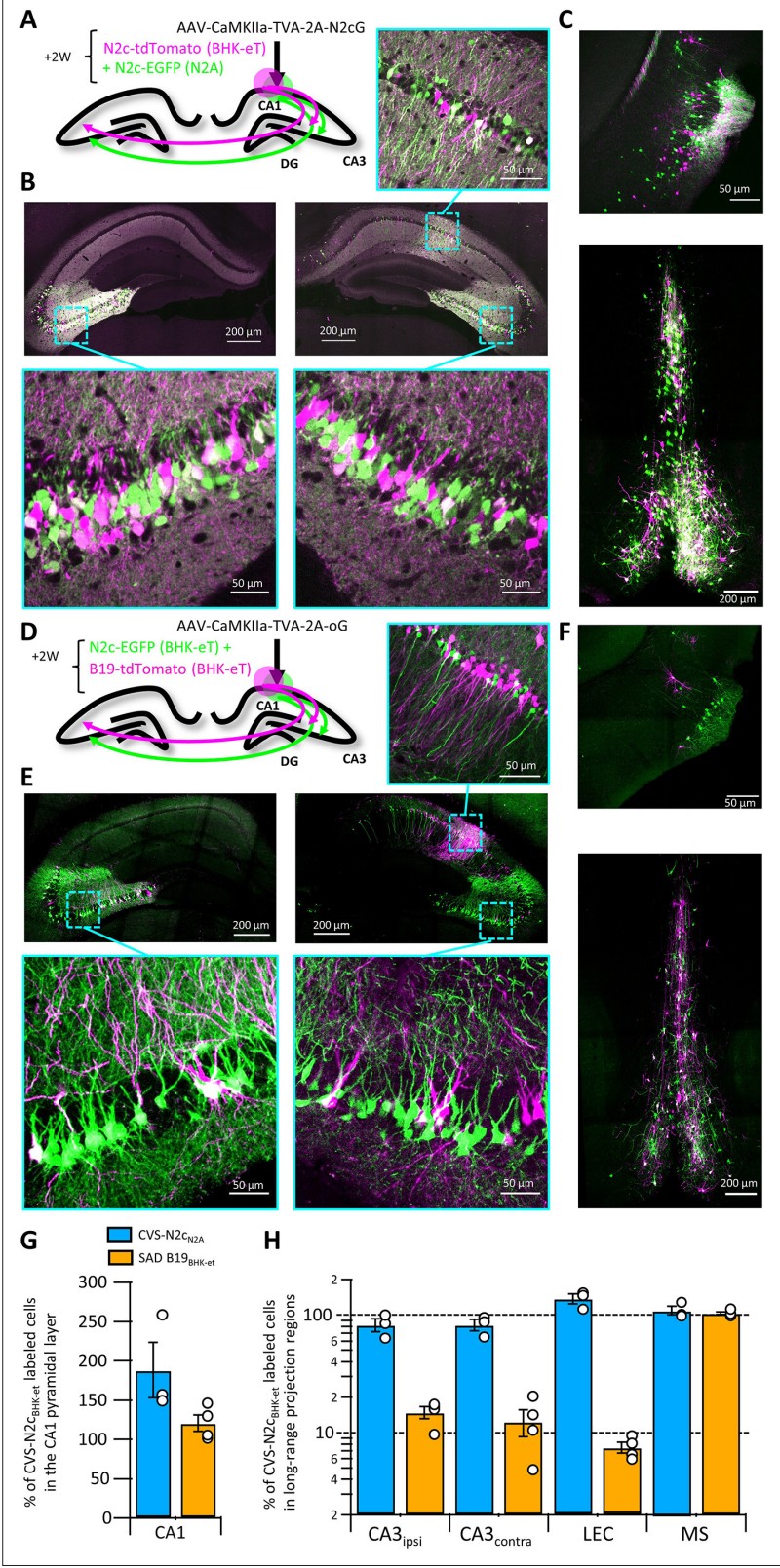

**Figure 3.** Retrograde labeling efficiency is determined by the viral strain, and not the packaging cell lines. (**A**) Schematic illustration of the injection scheme, designed for comparison of retrograde labeling efficacy between CVS-N2c vectors produced using either the N2a-based or BHK-based packaging cell lines, propagating using the N2c glycoprotein. (**B**) Representative confocal images of the ipsi- and contra-lateral hippocampus. Expanded

*Figure 3 continued on next page*

*Figure 3 continued*

images of the CA3 region of both hemispheres and the CA1 of the injection site correspond to the areas delineated by cyan rectangles. (**C**) Representative confocal images of the LEC (top) and the septal complex (bottom). (**D–F**) Same as (**A–C**) but for comparison of CVS-N2c and SAD B19 vectors, both produced with BHK-based packaging cell line and propagating using the SAD-B19 optimized glycoprotein (oG). (**G**) Summary bar plot showing the ratio of first order starter cells in the CA1 pyramidal layer, between neurons labeled with either CVS-N2c$_{N2a}$ (blue) or SAD B19$_{BHK-et}$ (orange) and the neurons labeled with CVS-N2c$_{BHK-et}$. (**H**) Summary bar plot showing the differences in retrograde labeling efficiency, under both injections schemes described in (**A**) and (**D**). All values were normalized to the ratio of starter cells shown in (**G**). N=4 and 3 animals for the N2c-N2c and N2c-B19 comparisons, respectively. Data shown as mean and SEM with black circles denoting individual animals.

The online version of this article includes the following source data and figure supplement(s) for figure 3:

**Source data 1.** Cell count for experimental designs and ratio calculations.

**Figure supplement 1.** Differential effects of N2c and B19 glycoproteins on propagation efficiency of CVS-N2c vectors.

labeling of neurons with N2c-tdTomato, whose somata were located in the *S.O.* and *S.LM.,* but not in the *Stratum Pyramidal* (*S.P.*) or *Radiatum* (*S.R.*) layers. In addition, we uncovered a third population of DG-projecting neurons in the superficial most layer of the subiculum (***Figure 4B***). Analysis of colocalization has revealed that while half of all retrogradely labeled neurons found in the *S.O* and *S.LM.* were also positive for EGFP, in the subicular population, which accounted for more than a third of all intrahippocampal projecting cells, colocalization was almost completely absent (***Figure 4C***), suggesting that this population consists of excitatory neurons. AAV vectors expressing EGFP under control of either the inhibitory neuron-specific mDLX or the excitatory neuron-specific CaMKIIa promoters, injected into the superficial CA1 or superficial subiculum, respectively, confirmed the existence of axonal branching in the DG (***Figure 4—figure supplement 1A*** and B). Retrograde labeling from adult-born DGCs or ventrally located DGCs revealed similar presynaptic populations (***Figure 4—figure supplement 1C*** and D), providing further support for the abundance of these connections.

Taking advantage of the high-throughput nature of our labeling approach, we immunostained sections from labeled animals for two of the most prominent interneuron markers, Parvalbumin (Pvalb) and Somatostatin (Sst), and found that while 43 ± 5% of *S.O.* retrogradely-labeled neurons were positive for Sst, none of the labeled cells in both *S.O.* and *S.LM* colocalized with Pvalb (***Figure 4D and E***). This is consistent with the known projection of *S.O.* Sst neurons to the *S.LM.*, which borders on the dentate gyrus, while axons of Pvalb neurons in the CA1 project mainly to the pyramidal cell layer and have few, if any axonal arborization in the *S.LM.* (***Freund and Buzsáki, 1996***) and are therefore much less likely to branch into the DG.

## Differential targeting of neuronal populations for retrograde labeling

The abundance of population-specific cre/flp driver mouse lines allows for a wide range of possible labeling experiments. However, often such lines are either unavailable, or insufficiently specific for the experimental requirements. To allow for a broader use of this tool, we have developed additional AAV vectors driving expression of the TVA-2A-N2cG cassette which can achieve greater specificity.

First, we wanted to be able to compare between two genetically distinct, yet spatially overlapping populations. To this aim, we designed AAV vectors which contain a cre-off mechanism, using a single-floxed, excisable open reading frame (SEO) to drive transgene expression under control of the excitatory neuronal CaMKIIa promoter (***Figure 5—figure supplement 1A***). We tested this tool on CA1 pyramidal neurons of *Calb1$^{cre}$* mice, in which Cre recombinase is expressed exclusively in deep, but not in superficial CA1 neurons (***Li et al., 2017***; ***Valero et al., 2015***; ***Figure 5A***). Parallel retrograde labeling from these two subpopulations of CA1 neurons revealed that while both receive relatively equal input from CA3 neurons, deep CA1 neurons receive substantially greater input from LEC-3 neurons (***Figure 5B***), confirming previous findings obtained using lower-throughput approaches (***Li et al., 2017***; ***Masurkar et al., 2017***).

Next, we wanted to test whether our vectors could also label projections to inhibitory neurons. To attain interneuron-specific retrograde labeling, we first used AAV vectors to express the TVA-2A-N2cG cassette under control of the interneuron-specific mDLX promoter (***Dimidschstein et al., 2013***) in the CA1 of GAD1-EGFP mice. A subsequent injection of CVS-N2c-tdTomato revealed widespread

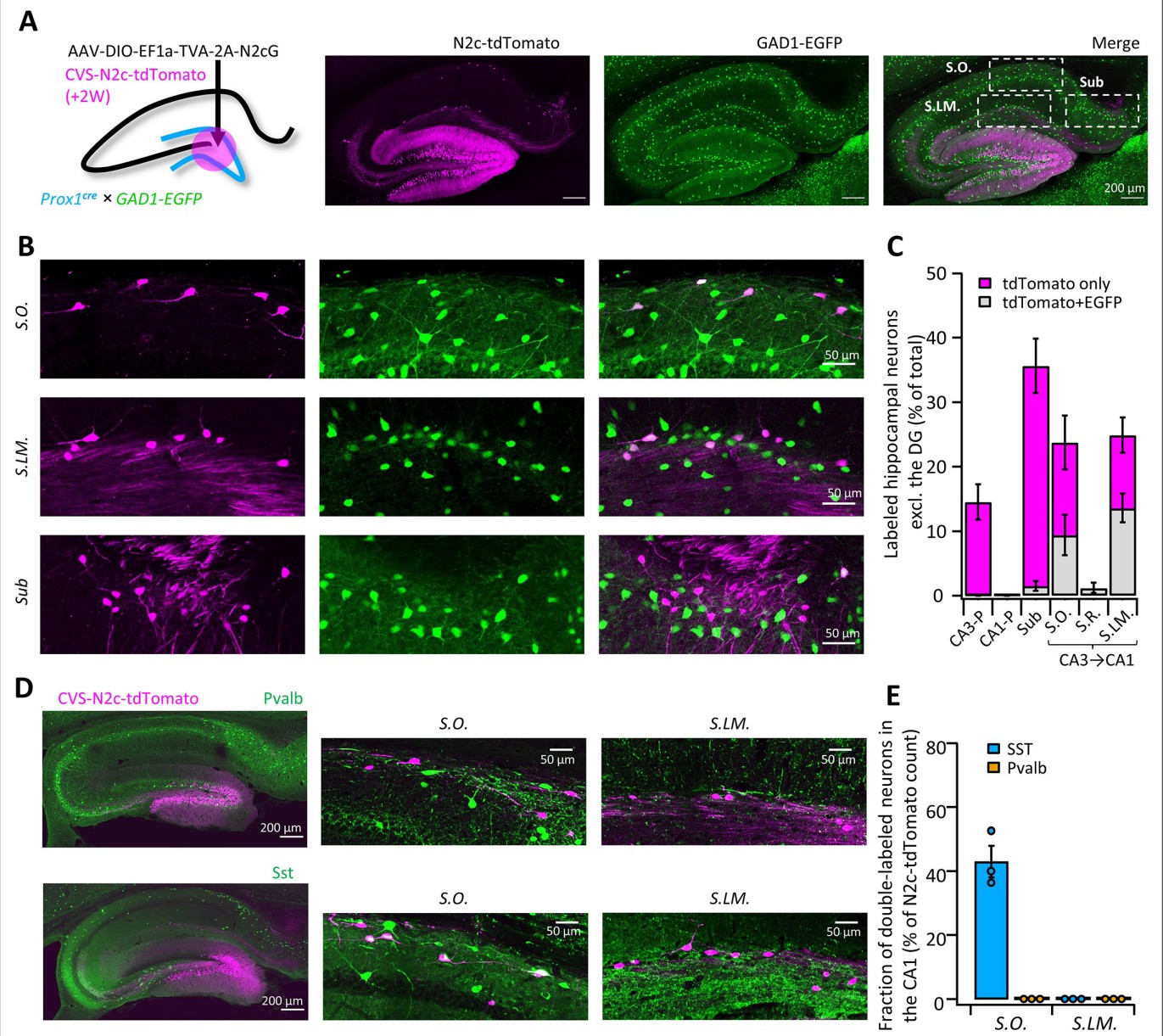

**Figure 4.** High-throughput retrograde labeling with CVS-N2c enables identification of non-canonical projections to DGCs. (**A**) Schematic illustration and representative confocal images, describing the injection scheme designed to target DGCs for retrograde labeling in an interneuron reporter line. (**B**) Representative confocal images (left) of the regions highlighted in (**A**) showing retrogradely-labeled neurons along specific hippocampal layers and their overlay with the interneuron-specific marker. (**C**) Summary bar plot showing the distribution of DG-projecting hippocampal neurons outside of the DG (magenta) and of them, the fraction of double-labeled neurons (grey). Calculation of cell numbers in the dendritic cell layers combined cells along the entire proximo-distal hippocampal axis, from CA3 to CA1. N=189 cells from 3 animals. (**D**) Representative parasagittal sections of the hippocampus following retrograde labeling from the DG with CVS-N2c-tdTomato, along with immunolabeling of parvalbumin (Pvalb, top) and Somatostatin (Sst, bottom). Expanded view of the *S.O.* and *S.LM.* are shown to the right of each image. (**E**) Summary plot describing the proportion of Pvalb- or Sst-positive neurons of the total CVS-N2c labeled neurons in the *S.O.* or *S.LM.* of the CA1. N=125 cells from three animals.

The online version of this article includes the following source data and figure supplement(s) for figure 4:

**Source data 1.** Cell count for experimental designs and ratio calculations.

**Figure supplement 1.** Cross validation of novel excitatory and inhibitory intrahippocampal projections to DGCs.

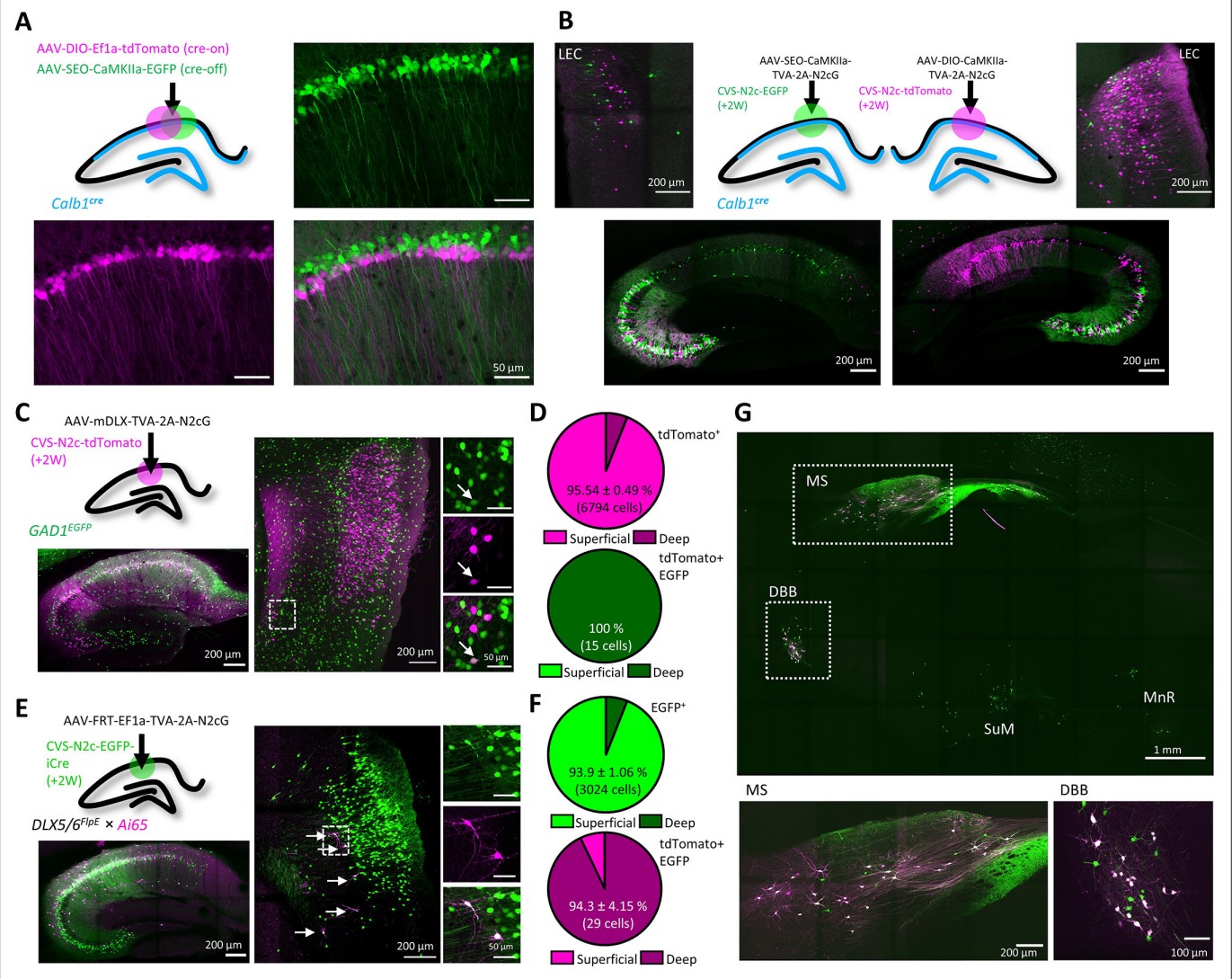

**Figure 5.** An AAV vector suite for targeting multiple and diverse neuronal populations. (**A**) Graphical representation (top left) and representative confocal images, demonstrating differential targeting of superficial and deep CA1 pyramidal neurons using a combination of cre-on and cre-off AAV vectors. (**B**) Graphical representation (top center) for dual retrograde labeling from superficial (bottom left, green) and deep (bottom right, magenta) CA1 pyramidal neurons and the resulting distribution pattern of their corresponding projection neurons in the EC (top left and top right). (**C**) Graphical representation of the viral injection scheme for mapping inputs into hippocampal inhibitory neurons (top left), and representative parasagittal images of labeled cells in the HC (bottom left) and LEC (right). Expanded images show a double-labeled neuron in EC-6. (**D**) Distribution of all retrogradely labeled cells (top) and of double-labeled cells only (bottom) among the superficial layers 2 and 3 and the deep layers 5 and 6 of the LEC. N=8 sections/3 animals. (**E**) Same as (**C**), but for Dlx5/6$^{FlpE}$ × Ai65 mice. (**F**) Same as (**D**) for the experiments described in (**E**). N=8 sections/3 animals. (**G**), A representative parasagittal image of deep brain structures following the injection scheme described in (**E**). SuM – Supramammilary Nucleus; MnR – Median nucleus Raphe.

The online version of this article includes the following figure supplement(s) for figure 5:

**Figure supplement 1.** Targeting specificity of newly designed AAV vectors.

labeling of 2$^{nd}$ order neurons both within the hippocampal CA1 and CA3 fields, as well as in the EC (***Figure 5C*** and ***Figure 5—figure supplement 1B***). An additional analysis of co-labeling in the EC revealed a small population of long-range inhibitory projection neurons, located preferentially in the deeper layers 5 a and 6 (***Figure 5D***), whose existence has previously been reported (***Basu et al., 2016***; ***Melzer et al., 2012***) but their location within the EC has so far remained unknown. In order to cross-validate and expand on these findings, we crossed Dlx5/6$^{flpE}$ mice with the double cre +flp, tdTomato reporter line Ai65 (***Madisen et al., 2015***). Hereby, using injections of AAV vectors with flp-dependent

TVA-2A-N2cG expression cassette, followed by RVdG$_{envA}$-CVS-N2c-EGFP-iCre, we were able to high-light and isolate the inhibitory projections to hippocampal inhibitory neurons, as the double cre +flp recombination required for tdTomato expression can only take place in inhibitory neurons transduced by the rabies virus (*Figure 5E*). In line with our previous findings, we show that while the majority of excitatory cortical input to hippocampal inhibitory neurons originated in the superficial layers 2 and 3, long-range inhibitory projection neurons are almost exclusively found in the deeper layers 5 a and 6 (*Figure 5F*). Further examination of labeling patterns of 2$^{nd}$ order neurons in deep-brain regions showed that in the MS and the diagonal band of Broca (DBB), a large fraction of cells are double-labeled, confirming that our labeling strategy for isolation of long-range inhibitory projections is exhaustive (*Figure 5G*). This strongly suggests that while the deeper cortical layers give rise to both inhibitory and excitatory hippocampal projections, the excitatory projection originated from a substantially larger population. This is particularly unexpected, as to-date, only sporadic evidence existed for the presence of cortico-hippocampal projections, excitatory or inhibitory, arising from the deeper layers of the EC (*Gloveli et al., 2001*).

## Bicistronic CVS-N2c vectors for efficient dual labeling

A previous report has shown that the B19 N-P linker sequence, which allows the virus to effectively separate these proteins, can also be used for separation of exogenous genes (*Osakada et al., 2018*). While this approach could promote an expansion of the rabies toolkit to accommodate more complex experimental designs, it remains to be shown to whether efficient separation indeed takes place and whether, unlike the use of 2 A peptides or the IRES sequence, expression levels of the individual proteins remain unaltered. To test the feasibility and efficacy of this approach, we designed new CVS-N2c bicistronic plasmids, to drive co-expression of a nuclear-localized EGFP (nl.EGFP) with either tdTomato (*Figure 6A*) or a synaptophysin-tethered EGFP for specific labeling of presynaptic terminals (SypEGFP, *Figure 6B*) separated by the CVS-N2c N-P liker sequence. Specific delivery of these vectors to hippocampal DGCs using targeted AAV expression of the cre-dependent TVA-2A-N2cG cassette in *Prox1$^{cre}$* mice, revealed that in both cases, the individual proteins were efficiently expressed in a compartment-specific manner (*Figure 6A and B* and *Figure 6—figure supplement 1A-D*).

Next, we capitalized on this result to create new bicistronic vectors for dual expression of the optogenetic actuator ChIEF (*Lin et al., 2009*), along with a fluorescent protein, speculating that the increased expression levels and the untethering of the fluorophore will lead to greater responsivity of labeled neurons to optogenetic stimulation, as well as facilitate their identification. We again targeted DGCs for retrograde labeling using these new vectors and recorded the action potential success rate of labeled neurons in the EC, following five light pulses, each 1 ms long, at varying frequencies. As predicted, action potentials could be reliably generated at much higher stimulation frequencies and at a much earlier time point following stimulation onset than has previously been shown for both SAD B19 and CVS-N2c vectors (*Osakada et al., 2018*; *Reardon et al., 2016*: *Figure 6C and D*). In these recorded neurons, the resting membrane potential was −55 mV to −75 mV, corroborating the functional integrity of the cells. From a total of 21 cells, 3 cells had lower membrane potential above −55 mV (but below −50 mV) and these were excluded from the final analysis. By recording synaptic responses to optogenetic stimulation from several neuronal populations sharing the same input as DGCs, such as neighboring, non-transduced DGCs, CA3 pyramidal neurons and dentate gyrus molecular layer interneurons, we also show that this tool can be reliably used for exploring circuit motifs, and also potentially for effective circuit-based manipulation in behaving animals (*Figure 6E*). Since our observations indicate that under in vitro conditions, cellular health and viability begin to deteriorate in preparations made >10 days from RVdG-CVS-N2c injection, it is important to show that effective optogenetic activation can be achieved at an earlier time point, when the cells remain healthy and viable for the duration of the experiment.

## In vivo measurements from extended cortical networks

While RVdG viral vectors are highly effective in describing circuit architecture, their neurotoxicity limits their use for many applications, in which long-term monitoring or manipulation of labeled circuit is required (*Luo et al., 2018*) and while a new technology for production of non-neurotoxic RVdG vectors was recently been presented (*Chatterjee et al., 2018*), the current inability to produce these vectors in a pseudotyped form excludes their use for cell-type-specific tracing experiments. RVdG-CVS-N2c

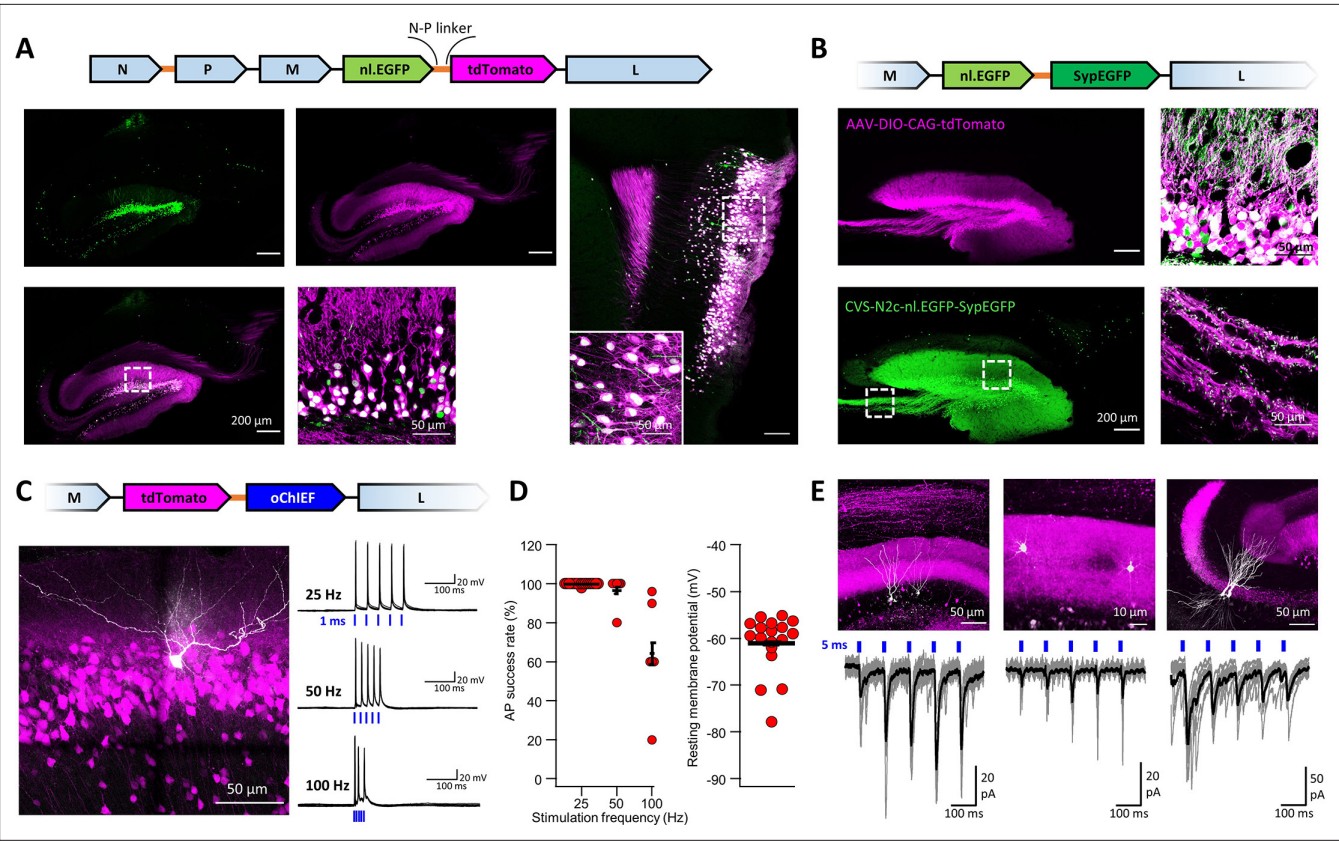

**Figure 6.** An extended suite of RVdG-CVS-N2c vectors for bicistronic expression of fluorescent markers and optogenetic effectors. (**A**) Schematic illustration of the vector sequence, designed to drive independent bicistronic expression of a nuclear-localized EGFP (nl.EGFP) alongside tdTomato, using the N-P linker sequence (top). Representative confocal images of the HC (bottom right) and EC (bottom left) following retrograde labeling from the DG, demonstrate the differential localization, indicating effective separation of the fluorophores. (**B**) Schematic diagram of a bicistronic nl.EGFP +SypGFP CVS-N2c vector (top) used for retrograde labeling from the DG (right panels) and representative confocal images demonstrating dual nuclear and synaptic localization of EGFP in the dentate granular and molecular layer (top right image) and purely synaptic localization at the mossy fibers terminals (bottom right image). (**C**) Schematic diagram of a bicistronic tdTomato +oChIEF CVS-N2c vector (top) used for retrograde labeling from the DG, and a representative image of a biocytin-filled neuron (white) in MEC-2 (bottom left) along with representative traces from 10 overlaid recordings at different frequencies (bottom right). (**D**) Summary plots of the action potential success rate for recordings made 6–7 days after introduction of RVdG (left) and their resting membrane potential at the time of recording (right) demonstrate the light responsiveness and physiological condition of the labeled neurons. (**E**) Representative confocal images (top) of DGCs (left), DG molecular layer interneurons (center) and CA3 pyramidal neurons (right) and their synaptic responses to optogenetic activation of the perforant path (bottom) following retrograde labeling from the dorsal DG with the bicistronic CVS-N2c-tdTomato-oChIEF vector.

The online version of this article includes the following source data and figure supplement(s) for figure 6:

**Source data 1.** Firing rate and membrane potential properties of neurons transduced with CVS-N2c-tdTomato-ChIEF, 7 days post transduction.

**Figure supplement 1.** Additional properties of bicistronic RVdG-CVS-N2c vectors.

vectors have previously been shown to be less neurotoxic than the SAD B19 strain, and compatible for prolonged imaging of neuronal activity in vivo (*Reardon et al., 2016*). However, since neurotoxicity has not been completely eliminated, it is still possible that their endogenous activity patterns become impaired within this time period, thereby impinging on results.

To address this question, we dissected a microcircuit within the primary visual cortex (V1) for in vivo calcium imaging, by first injecting retrogradely transported Cre-expressing AAV vectors (*Tervo et al., 2016*) in the laterodorsal nucleus of the thalamus (LD) and AAV-DIO-CAG-tdTomato+AAV-DIO-EF1a-TVA-2A-N2cG into the V1. A subsequent injection of RVdG$_{envA}$-CVS-N2c-GCaMP8m into V1 resulted in specific retrograde labeling from layer 5 neurons in the V1 (*Figure 7A and B* and *Figure 7—figure*

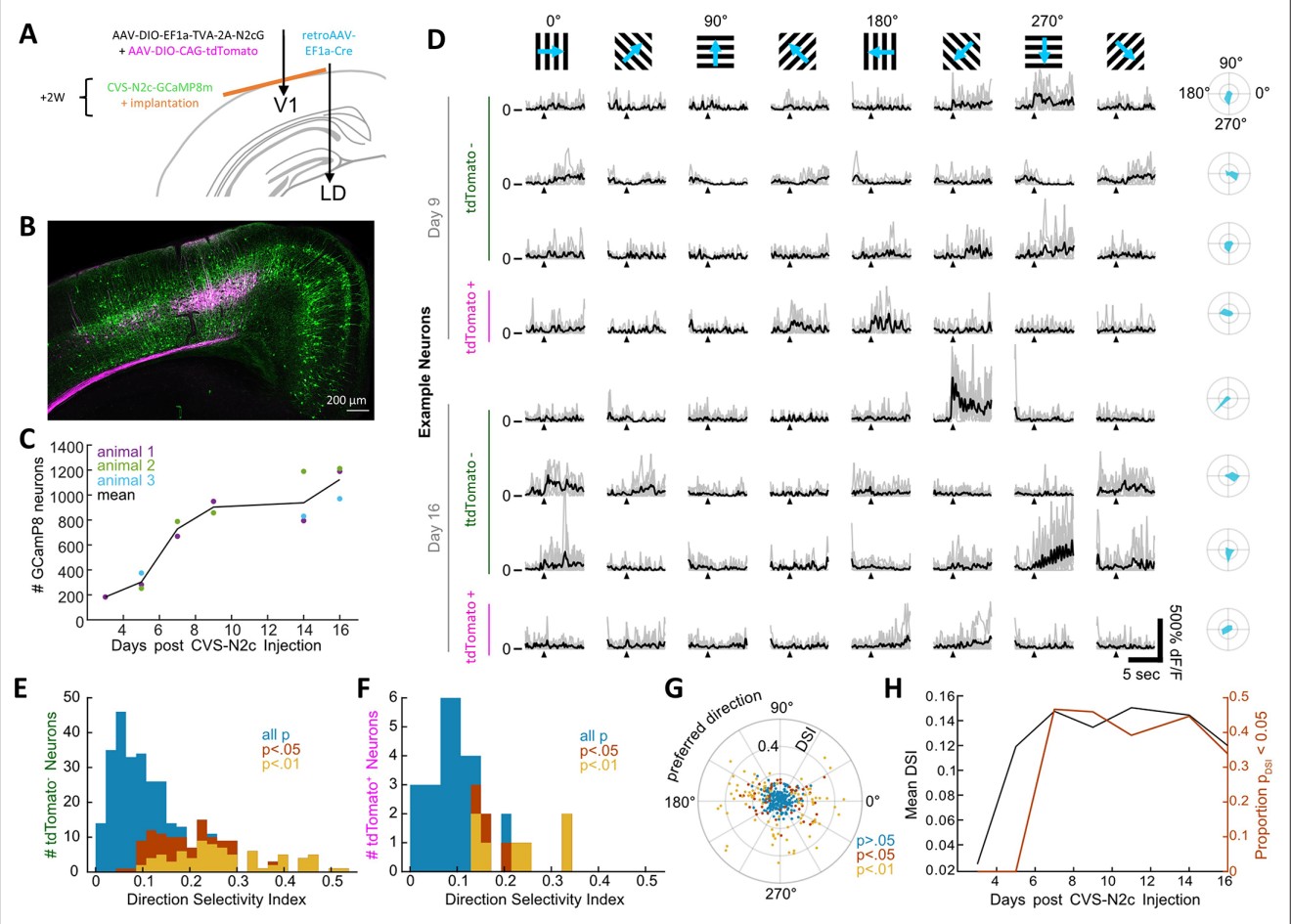

**Figure 7.** New CVS-N2c vectors for in vivo two-photon long-term imaging of activity in a cortical microcircuit. (**A**) Illustration of the injection scheme for labeling projections onto the V1 region's layer 5 neurons with GCaMP8m. (**B**) A representative confocal image of a coronal section following the injection scheme described in (**A**). (**C**) GCaMP8m-positive neuron numbers over recording days. (**D**) In vivo two-photon imaging in presynaptic neurons. Drifting gratings in 8 directions (top row). Trial responses (grey) and average (black) for example neurons on day 9 and day 16, stimulation starts at black triangle. Polar plot of the directional responses in right column. (**E**) Histogram of direction selectivity for tdTomato-negative (2nd order) neurons. (**F**) Same as in E for tdTomato-positive (starter) neurons. (**G**) Polar scatter plot of direction selectivity (radial) over preferred direction (angular) for all recorded neurons. (**H**) Mean DSI and proportion of significant direction selectivity over recording days.

The online version of this article includes the following source data and figure supplement(s) for figure 7:

**Source data 1.** Calcium transient properties of GCaMP8m expressing neurons in vivo.

**Figure supplement 1.** Retrograde labeling specificity for in vivo calcium imaging with RVdG$_{envA}$-CVS-N2c-GCaMP8m.

*supplement 1A* and B). In the following weeks, we observed the time course of GCaMP8m expression and recorded calcium transients of the labeled 1st and 2nd order neurons using two-photon imaging, in response to visual cues in three awake behaving mice. While we detected the first GCaMP8m labeled neurons already at day 3, substantial labeling at approximately half-maximal numbers in all animals started 7 days after RVdG$_{envA}$-CVS-N2c-GCaMP8m injection. Subsequently, the increase of neuronal numbers slowed and remained high until day 16 (on average 1100 neurons), when the experiments were terminated (*Figure 7C*). While the most superficial tdTomato-labeled starter neurons were detectable in vivo, their depth did not allow a robust estimate of their total numbers in our configuration.

Each session, we recorded fluorescence time series for a subset of the GCaMP8m-labeled neurons and determined their calcium activity while presenting drifting grating stimuli to the mouse in a visual dome setup (*Figure 7D*). Both tdTomato-positive (putative 1st order) neurons as well as tdTomato-negative 2nd order neurons showed similar tuning properties (*Figure 7E and F*) with preferred directions slightly biased in the horizontal plane (*Figure 7G*). The fluorescence signal-to-noise ratio per

neuron (*Figure 7—figure supplement 1C*) and the proportion of direction tuned neurons (*Figure 7H*) remained constant from day 7 to the conclusion of the experiments at day 16. These experiments demonstrate that RVdG-CVS-N2c viral vectors can be reliably used for in vivo experiments in behaving animals for circuit-specific recording and manipulation of activity and delineate a time period during which the neurons retain their normal physiological properties.

## Discussion

### Fast and efficient production of high-titer rabies virus vectors

The published production protocol for RVdG$_{envA}$-CVS-N2c rabies viral vectors utilizes the neural precursor N2A cell line since it was hypothesized to increase neuronal tropism, thereby increasing transsynaptic spread from the starter cells. However, since this cell line is relatively more difficult to transfect and maintain than other cell lines, this choice resulted in a lengthy and cumbersome production process, with low titer yields (*Reardon et al., 2016*). As this premise has not been directly tested, we surmised that the increased labeling efficiency stems from the N2c strain itself, rather than its production process, and created an alternative packaging system to optimize all the different steps of production. By co-expressing mammalian-optimized versions of the genes required for the rabies amplification process, in tandem with antibiotics-resistance genes driven by strong constitutive promoters, in highly resilient and easily maintained cell lines, our method effectively minimizes production time, while simultaneously increasing viral titers by orders of magnitude. The additional co-expression of TVA and envA in the pseudotyping cell lines reduces to a minimum the amount of native-coat virus needed to initiate the process, thereby nearly abolishing the presence of these particles in the pseudotyped stock, and vastly increasing tracing specificity. This advantage is particularly relevant when performing retrograde labeling from single cells (*Rossi et al., 2020*; *Wertz et al., 2015*; *Wickersham et al., 2007b*) where a small number of non-specifically labeled neurons can potentially account for a large fraction of the entire labeled population.

RVdG-CVS-N2c vectors have previously been shown to outperform the SAD B19 in almost any parameter measured (*Ohara et al., 2018*; *Reardon et al., 2016*), attributed to the use of neural precursor cells for their production. We show that contrary to the premise, vectors produced using the HEK-GT/BHKeT packaging system show comparable retrograde labeling efficiency to ones produced in N2A cells and significantly higher than that of SAD B19 particles. This attribute has allowed us to efficiently find and characterize non-canonical hippocampal connections, for which only anecdotal reports have been available. While we show that neurons transduced with CVS-N2c vectors produced using our cell lines maintain viability and physiological function for at least 16 days, since no direct comparisons of physiological properties and long-term viability were made with vectors produced in the N2A line, it remains possible that packaging these vectors in N2A cells would result in reduced neurotoxicity, when compared with vector packaged in non-neuronal cell lines.

Using these new vectors, we were able to demonstrate the ease in which non-canonical projections to a target population can be teased out. We chose to focus these efforts in the DG, as the ample existing anecdotal evidence allowed us to compare our results against previously verified data (*Hájos and Mody, 1997*; *Katona et al., 2017*; *Klausberger and Somogyi, 2008*; *Szabo et al., 2017*). In line with these reports, retrograde labeling from DGCs labeled two distinct populations of inhibitory neurons, residing in the *S.O.* and *S.LM* of the CA1, which mainly contain neurons projecting to the apical dendrites of CA1 neurons. In addition to these aforementioned populations, we have also observed a third population of labeled neurons along the superficial most layer of the subiculum, directly in the path of perforant path fibers. Unlike the previous ones, these cells did not express the inhibitory fluorescent indicator for GAD1, which suggests that this might be a yet undiscovered intra-hippocampal excitatory projection to the DG. Since in the GAD1-EGFP line not all inhibitory neurons may express the fluorescent marker (*Tamamaki et al., 2003*), it remains a possibility that these neurons are also inhibitory but are somehow genetically indisposed to express the marker. However, their pyramidal-like morphology, spiny dendritic arbors and putative expression of the CaMKIIa promoter, all hallmarks of excitatory neurons, render this possibility unlikely.

The successful deployment of RVdG$_{envA}$ viral vectors is mainly limited by the ability to genetically dissect the target population. While many different mouse lines have been previously used for this purpose, covering the vast majority of genetically unique populations, the robust amplification

capabilities of RVdG-CVS-N2c vectors require a high degree of specificity, in order to avoid possible off-target effects. We have developed several approaches that could further improve experimental paradigms, designed to attain results that are more specific. For example, cre-off viruses with specific promoters for AAV vectors can be used together with their cre-on equivalents in order to determine which projections are specific to a target population in a given region, and which are shared by the general population of neurons in a given brain region. In addition, intersectional genomic and viral-borne recombination could be used to dissect and highlight specific sub-populations of projection neurons. Using the vectors we designed, this toolbox can be expanded further to include other restriction approaches, such as tet-controlled elements.

Complementing these tools is a new suite of mono- and bicistronic RVdG-CVS-N2c vectors, expressing a broad range of fluorophores, recombinases, synaptic markers, optogenetic actuators and genetically encoded calcium indicators, to enable for diverse experimental purposes (see Key resources table). We demonstrate the flexibility of this approach, by expressing and imaging the recently developed calcium indicator GCaMP8m (*Zhang et al., 2021*) in the presynaptic neuronal population of starter neurons, that themselves project to the thalamus. Furthermore, we show that using the CVS-N2c endogenous N-P linker, we could effectively separate at least two individual elements, and possibly more. Apart from enabling better tracking of distinct subcellular compartments, we also show that the separation of the fluorophore from optogenetic actuators can increase their light responsiveness, possibly as a result of improved membrane trafficking and higher expression levels. However, some of these properties should also be attributed to the improved ChR2 variant we used, which was shown to possess faster kinetics (*Lin et al., 2009*). Another benefit of this separation is better identification of labeled neurons, since the untethering of the fluorophore from membrane-bound proteins leads to its predominantly cytosolic expression.

## Conclusion

We present here a comprehensive toolkit for rapid, high-throughput production of RVdG-CVS-N2c rabies viral vectors, along with an extended multipurpose AAV and CVS-N2c vector suites. These vectors now allow extensive labelling of presynaptic projections, facilitating a more comprehensive understanding of neuronal network wiring, at greater specificity, efficiency and versatility for the experimenter. Together, this system enables a fast, simple and highly effective approach for circuit mapping in the mammalian brain.

# Materials and methods

**Key resources table**

| Reagent type (species) or resource | Designation | Source or reference | Identifiers | Additional information |
|---|---|---|---|---|
| Strain, strain background (mouse Prox1$^{cre}$) | Tg(Prox1-cre)SJ32Gsat/Mmucd | MMRRC (N. Heintz) | 036644-UCD | |
| Strain, strain background (mouse Calb1$^{cre}$) | B6;129S-*Calb1$^{tm2.1(cre)Hze}$*/J | Jackson labs (H. Zeng) | 028532 | |
| Strain, strain background (mouse Ascl1$^{creERT2}$) | *Ascl1$^{tm1.1(Cre/ERT2)Jejo}$*/J | Jackson labs (J. Johnson) | 012882 | |
| Strain, strain background (mouse Ai14) | B6.Cg-Gt(ROSA)26Sor$^{tm14(CAG-tdTomato)Hze}$/J | Jackson labs (H. Zeng) | 007914 | |
| Strain, strain background (mouse GAD1$^{EGFP}$) | Not deposited | K.Obata and Y.Yanagawa Tamamaki, N., Yanagawa, Y., Tomioka, R., Miyazaki, J.I., Obata, K., and *Tamamaki et al., 2003*. Green fluorescent protein expression and colocalization with calretinin, parvalbumin, and somatostatin in the GAD67-GFP knock-in mouse. Journal of Comparative Neurology 467, 60–79. https://doi.org/10.1002/cne.10905. | | |

*Continued on next page*

*Continued*

| Reagent type (species) or resource | Designation | Source or reference | Identifiers | Additional information |
|---|---|---|---|---|
| Strain, strain background (mouse Dlx5/6[FlpE]) | Tg(mI56i-flpe)39Fsh/J | Jackson labs (G. Fishell) | 010815 | |
| Strain, strain background (mouse RCE-FRT) | Gt(ROSA)26Sor[tm1.2(CAG-EGFP)Fsh]/Mmjax | Jackson labs (G. Fishell) | 32038 | |
| Strain, strain background (mouse Ai65) | B6;129S-*Gt(ROSA)26Sor*[tm65.1(CAG-tdTomato)Hze]/J | Jackson labs (H. Zeng) | 010815 | |
| Cell line (human) | HEK293T | ATCC | CRL-3216 | |
| Cell line (hamster) | BHK-21 | ATCC | CCL-10 | |
| Cell line (human) | HEK-GT | This paper | | |
| Cell line (human) | HEK-TVA | This paper | | |
| Cell line (hamster) | BHK-eT | This paper | | |
| Recombinant DNA reagent (plasmid) | pCAG-B19N | AddGene (I. Wickersham) | #59924 | |
| Recombinant DNA reagent (plasmid) | pCAG-B19P | AddGene (I. Wickersham) | #59925 | |
| Recombinant DNA reagent (plasmid) | pCAG-B19L | AddGene (I. Wickersham) | #59922 | |
| Recombinant DNA reagent (plasmid) | pAdDeltaF6 | AddGene (J. Wilson) | #112867 | |
| Recombinant DNA reagent (plasmid) | rAAV-DJ RepCap | Mark A. Kay | | |
| Recombinant DNA reagent (plasmid) | rAAV2-retro helper | AddGene (A. Karpova and D. Schaffer) | #81070 | |
| Recombinant DNA reagent (plasmid) | pAAV-EF1a-Cre | AddGene (K. Deisseroth) | #55636 | |
| Recombinant DNA reagent (plasmid) | pAAV-DIO-hSyn-mCherry | AddGene (K. Deisseroth) | #114472 | |
| Recombinant DNA reagent (plasmid) | RVdG-RVDG-CVS-N2c-EGFP | AddGene (T. Jessell) | #73461 | |
| Recombinant DNA reagent (plasmid) | RVdG-RVDG-CVS-N2c-tdTomato | AddGene (T. Jessell) | #73462 | |
| Recombinant DNA reagent (plasmid) | pAAV-DIO-Ef1a-TVA-2A-oG | This paper | #172359 | |
| Recombinant DNA reagent (plasmid) | pAAV-DIO-Ef1a-TVA-2A-N2cG | This paper | #172360 | |
| Recombinant DNA reagent (plasmid) | pAAV-FRT-EF1a-TVA-2A-N2cG | This paper | #172361 | |
| Recombinant DNA reagent (plasmid) | pAAV-DIO-CaMKII-TVA-P2A-N2cG | This paper | #172362 | |
| Recombinant DNA reagent (plasmid) | pAAV-SEO-CaMKII-TVA-P2A-N2cG | This paper | #172363 | |
| Recombinant DNA reagent (plasmid) | pAAV-mDLX-TVA-2A-N2cG | This paper | #172364 | |
| Recombinant DNA reagent (plasmid) | pAAV-DIO-mDLX-TVA-2A-N2cG | This paper | #172365 | |

*Continued*

| Reagent type (species) or resource | Designation | Source or reference | Identifiers | Additional information |
|---|---|---|---|---|
| Recombinant DNA reagent (plasmid) | pAAV-DIO-CAG-TVA-P2A-dTomato | This paper | #177016 | |
| Recombinant DNA reagent (plasmid) | pAAV-DIO-EF1a-TVA-P2A-EYFP | This paper | #177017 | |
| Recombinant DNA reagent (plasmid) | pAAV-SEO-CaMKII-EGFP | This paper | #177018 | |
| Recombinant DNA reagent (plasmid) | MMLV-CAG-TVA-IRES-Puro | This paper | #172366 | |
| Recombinant DNA reagent (plasmid) | MMLV-CAG-SADB19_oG-IRES-Puro | This paper | #172367 | |
| Recombinant DNA reagent (plasmid) | MMLV-CAG-G_oT7pol-IRES-BSD | This paper | #172369 | |
| Recombinant DNA reagent (plasmid) | pLV-EF1a-N2c_envA-IRES-Neo | This paper | #172368 | |
| Recombinant DNA reagent (plasmid) | RVDG-CVS-N2c-tdTomato-ChIEF | This paper | #172370 | |
| Recombinant DNA reagent (plasmid) | RVDG-CVS-N2c-EGFP-ChIEF | This paper | #172371 | |
| Recombinant DNA reagent (plasmid) | RVDG-CVS-N2c-EGFP-iCre | This paper | #172372 | |
| Recombinant DNA reagent (plasmid) | RVDG-CVS-N2c-EGFP-FlpO | This paper | #172373 | |
| Recombinant DNA reagent (plasmid) | RVDG-CVS-N2c-tdTomato-iCre | This paper | #172374 | |
| Recombinant DNA reagent (plasmid) | RVDG-CVS-N2c-tdTomato-FlpO | This paper | #172375 | |
| Recombinant DNA reagent (plasmid) | RVDG-CVS-N2c-mTurquoise | This paper | #172376 | |
| Recombinant DNA reagent (plasmid) | RVDG-CVS-N2c-E2_Crimson | This paper | #172377 | |
| Recombinant DNA reagent (plasmid) | RVDG-CVS-N2c-nl.mCherry-FlpO | This paper | #172378 | |
| Recombinant DNA reagent (plasmid) | RVDG-CVS-N2c-nl.EGFP-FlpO | This paper | #172379 | |
| Recombinant DNA reagent (plasmid) | RVDG-CVS-N2c-nl.EGFP-SypGFP | This paper | #172380 | |
| Recombinant DNA reagent (plasmid) | RVDG-CVS-N2c-SypRFP | This paper | #172381 | |
| Recombinant DNA reagent (plasmid) | RVDG-CVS-N2c-nl.EGFP-tdTomato | This paper | #172382 | |
| Recombinant DNA reagent (plasmid) | RVDG-CVS-N2c-EYFP | This paper | #172383 | |
| Recombinant DNA reagent (plasmid) | RVDG-CVS-N2c-mCitrine | This paper | #172384 | |
| Recombinant DNA reagent (plasmid) | RVDG-CVS-N2c-nl.mCherry-GCaMP7s | This paper | #172385 | |
| Recombinant DNA reagent (plasmid) | RVDG-CVS-N2c-nl.EGFP-jRGECO1a | This paper | #172386 | |

*Continued on next page*

*Continued*

| Reagent type (species) or resource | Designation | Source or reference | Identifiers | Additional information |
|---|---|---|---|---|
| Recombinant DNA reagent (plasmid) | RVDG-CVS-N2c-GCaMP8f | This paper | #172387 | |
| Recombinant DNA reagent (plasmid) | RVDG-CVS-N2c-GCaMP8m | This paper | #172388 | |
| Recombinant DNA reagent (plasmid) | RVDG-CVS-N2c-GCaMP8s | This paper | #172389 | |
| Antibody | Anti parvalbumin (rabbit polyclonal) | Swant antibodies | PV-27 | 1:1000 dilution |
| Antibody | Anti somatostatin (rabbit polyclonal) | BMA Biomedicals | T-4102 | 1:1000 dilution |
| Antibody | Anti EGFP (Chicken polyclonal) | Abcam | AB13970 | 1:1000 dilution |
| Antibody | Anti FLAG (mouse monoclonal) | Sigma Alderich | F1804 | 1:1000 dilution |
| Antibody | Alexa Fluor 647-conjugated goat anti-rabbit | Invitrogen | **A-21244** | 1:1000 dilution |
| Antibody | Alexa Fluor 647-conjugated goat anti-mouse | Invitrogen | **A-21235** | 1:1000 dilution |
| Antibody | Alexa Fluor 488-conjugated goat anti-chicken | Invitrogen | A-11039 | 1:1000 dilution |

## Generation of stable packaging cell lines

Retro-, and lentiviral vectors were produced by transfecting HEK293T cells (CRL-3216, ATCC) with one of the following vectors: pMMLV-CAG-SAD B19_optimized_G-IRES-Puro, pMMLV-CAG-optimized_T7pol-IRES-BSD, pLenti-EF1a-envA-IRES-Neo and pMMLV-CAG-TVA-IRES-Puro, along with the compatible retro- or lenti-viral GAG-Pol and the vesicular stomatitis virus glycoprotein (VSV-G), by means of calcium-phosphate precipitation. Viruses were collected and filtered 48 hr post transfection and used to transduce low-passage HEK293T or BHK-21 (CCL-10, ATCC) cells. Three days post transduction, the cells were passaged and one of the following antibiotics were added to the medium in order to select for the cells which stably express the respective construct: Puromycin dihydrochloride (3 µg ml$^{-1}$), blasticidine S hydrochloride (15 µg ml$^{-1}$), or G418 disulfate (Neomycin, 500 µg ml$^{-1}$, Sigma-Aldrich in all cases). Once the cells reached full confluence again, they were passaged, and again supplemented with the antibiotics. This cycle was repeated for at least three times a week for two weeks before initial use for production of rabies viral vectors. The new cell lines have all tested negative for mycoplasma. All cell lines and plasmids presented in this study are available from the corresponding author upon request and all plasmids are also available from Addgene (See Key resources table).

## Generation of bicistronic CVS-N2c rabies viral vectors

A method to produce bicistronic reading frames in rabies viral vectors has previously been described for vectors of the SAD B19 strain (*Osakada and Callaway, 2013*). Here, we adapted this approach for CVS-N2c vectors by inserting the virus's endogenous N-P linker between two coding sequences, which led to efficient separation of the proteins with no observable effect on the expression levels of either. The sequence for the linker used is: CATGAAAAAActAACACTCCTCC (lower case letters indicate the N-P boundary).

## Production of rabies viral vectors

HEK293-GT cells were used to rescue both SAD B19 and CVS-N2c rabies viral vectors. First, the cells were plated in a 35 mm culture dish and allowed to grow till they reached 80–90% confluence. Subsequently, they were transfected with the rabies vector plasmid and the SADB19 helper plasmids pTIT-N, pTIT-P and pTIT-L using polyethylenimine (PEI). Twenty-four hours later, the transfected cells were resuspended and re-plated in a 100 mm culture dish and incubated at 37 °C/5% CO$_2$ until they regained full confluence. Cells were maintained that way with frequent medium changes until ~100% of the cells were fluorescent, usually 5–6 days from time of transfection and 1–2 days from the point

fluorescence was first detected. At this point the medium was harvested, filtered, aliquoted, and kept at −80 °C until further use.

For pseudotyping of rabies vectors, BHK-eT cells were used to simultaneously pseudotype and amplify the vectors: First, low confluence BHK-eT cells were plated in two 100 mm culture dishes and each transduced with 0.5 ml of the native-coat virus. Once the cells reached full confluence, they were washed twice with Dulbecco's modified Eagle medium (DMEM), resuspended and each re-plated in a new 150 mm culture dish. Once the cells reached full confluence again and ~100% of them were fluorescent (~3 days post transduction), the medium was collected, filtered, stored at 4 °C and replaced with fresh medium. This process was then repeated for two to three consecutive days. Following the last collection, the virus was pooled and centrifuged at 70,000 rcf for 1.5 hr. Following centrifugation, the medium was aspirated and the viral pellet resuspended in 200 µl phosphate-buffered saline (PBS), pH 7.4, aliquoted and stored at −80 °C until use.

For titration of envA-pseudotyped rabies viral vectors, HEK293-TVA were plated in 35 mm wells at low confluence along with one well containing HEK293T cells for detection of unpseudotyped particles. The following day, cells from one of the HEK293-TVA wells were resuspended and counted, while the cells in the remaining wells were transduced with 1 µl concentrated virus in serial dilutions ranging from 1:10–1:10,000. To estimate the presence of native coat particles in each preparation, HEK293T cells in similar confluence were transduced in parallel with 1 µl of undiluted virus. Three days later, the cells were resuspended, washed with PBS and fixed with 4% paraformaldehyde (PFA). The fraction of transduced cells in each well was determined using flow cytometry (FACS Aria III), where the most extreme cell in a population of non-treated cells was used to determine the threshold. The titer was finally calculated based on a previously published formula (*Wickersham et al., 2010*). In all of the titrations performed for viruses produced using our packaging system fluorescently labeled cells were not detected in the control plate containing HEK293T cells, even when transduced with vectors at concentrations an order of magnitude higher than required for complete labeling of HEK-TVA cells. While on some occasions, cells in this control condition were detected beyond the predetermined threshold, these always had substantially weaker fluorescence than the peak fluorescence of the positively labeled cells. While it cannot be completely ruled out that these originate from cells transduced with native coat particles, it is more likely that this weak signal originates from slightly higher autofluorescence.

## Production of adeno-associated viral vectors

Adeno-associated virus (AAV) production was performed in HEK293T cells based on a previously-published protocol (*McClure et al., 2011*). Briefly, fully confluent HEK293 cells were transfected with an AAV2 vector plasmid along with pAdenoHelper and the AAV-dj RepCap plasmids using PEI. Thirty-six hours post transfection, the cells were harvested, pelleted, and lysed using three freeze-thaw cycles. The lysed cells were incubated with benzonase-nuclease (Sigma-Aldrich) for one hour and then the debris was pelleted and the virus-containing supernatant collected and passed through a 0.22 µm filter. The collected supernatant was subsequently mixed with an equal amount of heparin-agarose (Sigma-Aldrich) and kept at 4 °C overnight with constant agitation. The following day, the agarose-virus mixture was transferred to a chromatography column and the agarose was allowed to settle. The supernatant was then drained from the column by means of gravity and the agarose-bound virus was washed once with PBS and then eluted using PBS supplemented with 0.5 M NaCl. The eluted virus was then filtered again, desalinated and concentrated using a 100 kDa centrifugal filter and then aliquoted and stored at −80 °C until use. All AAV plasmids presented in this study are available from Addgene (see Key resources table).

## Animals

All transgenic driver and reporter lines have been previously characterized(see Key resources table). In all experiments, male and female mice were used interchangeably in equal proportions, in an age range which varied between 1 and 6 months old. Neither sex nor age-related differences could be observed in any of the measurements. Experiments on C57BL/6 wild-type and transgenic mice were performed in strict accordance with institutional, national, and European guidelines for animal experimentation and were approved by the Bundesministerium für Wissenschaft, Forschung und Wirtschaft

and Bildung, Wissenschaft und Forschung, respectively, of Austria (A. Haslinger, Vienna; BMWF-66.018/0010-WF/V/3b/2015; BMBWF-66.018/0008-WF/V/3b/2018).

## Organotypic hippocampal slice culture preparation

Hippocampal organotypic slice cultures were prepared from both hemispheres using the interface method (*Stoppini et al., 1991*). The entire hippocampus with entorhinal cortex was dissected from the brain of 5- to 8-day-old wild type mouse pups and cut perpendicularly to the longitudinal axes using a McIllwain tissue chopper. Hippocampus extraction and cutting were performed in a medium containing Hanks' Balanced Salt Solution (HBSS, Gibco) and 20% D-glucose (Braun). Slices were placed on microporous membrane inserts (Millicell, Millipore) floating on culture media containing 50% minimum essential medium, 25% basal medium Eagle, 25% heat-inactivated horse serum, 2 mM glutamax (all from GIBCO) and 0.62% D-glucose (Braun). Slice cultures were kept at 37 °C and 5% $CO_2$, until used for viral transduction experiments.

## Stereotaxic intracranial virus injections

For in vivo delivery of viral vectors, 1- to 6-months-old male or female mice were anesthetized with isoflurane, injected with analgesics and placed in a stereotaxic frame where they continued to receive 1–5% isoflurane vaporized in oxygen at a fixed flow rate of 1 l $min^{-1}$. Leg withdrawal reflexes were tested to evaluate the depth of anesthesia and when no reflex was observed an incision was made across the scalp to expose the skull. Bregma was then located, and its coordinates used as reference for anterior-posterior (AP) and medio-lateral (ML) coordinates while the surface of the dura at the injection site was used as reference for dorso-ventral (DV) coordinates. In our experiments, we used the following sets of AP/ML/DV coordinates (in mm): DG: −1.9/1.3/−1.9; CA1: −1.9/1.5/−1.2; V1: −3.5/2/−0.7; LD: −1.3/1.3/−2.5. AAV vectors were first diluted 1:5 in PBS and delivered to the injection site at a volume of 0.3 μl and a rate of 0.06 μl $min^{-1}$, using a Hamilton syringe and a 32 G needle. After the injection was completed, the needle was left in place for a few additional minutes to allow the virus to diffuse in the tissue and then slowly retracted. At the end of the injection session, the scalp was glued back together, and the mice were returned to their home cage to recover. Injection of pseudotyped rabies viral vectors took place 2–3 weeks after initial injection of AAV vectors containing the TVA receptor and rabies glycoprotein. Pseudotyped rabies vectors were first diluted to reach a final concentration of ~2–5 ×$10^8$ TU $ml^{-1}$ and then injected in the same manner as the AAV. Except for rabies injections into the DG of Prox1-cre transgenic animals, all other injections of rabies virus were shifted −0.2 mm AP and −0.2 mm ML. This was done in order to avoid, as much as possible, non-specific labeling along the needle tract of the first injection, due to the lack of complete specificity of cre-recombinase expression in the other transgenic lines used in this study.

## Slice preparation and electrophysiology

Electrophysiological recordings from identified retrogradely-labeled cells were performed 5–7 days following injection of CVS-N2c vectors. Manipulated animals were anaesthetized using an MMF mixture consisting of medetomidin (0.5 mg $kg^{-1}$), midazolam (5 mg $kg^{-1}$) and fentanyl (0.05 mg $kg^{-1}$) and subsequently perfused through the heart with 20 ml ice-cold dissection solution containing 87 mM NaCl, 25 mM $NaHCO_3$, 2.5 mM KCl, 1.25 mM $NaH_2PO_4$, 10 mM D-glucose, 75 mM sucrose, 0.5 mM $CaCl_2$, and 7 mM $MgCl_2$ (pH 7.4 in 95% $O_2$/5% $CO_2$, 325 mOsm). The brain was then removed and the hippocampus along with the adjacent cortical tissue was dissected out and placed into a precast mold made of 4% agarose designed to stabilize the tissue. The mold was transferred to the chamber of a custom-built or a VT1200 vibratome (Leica Microsystems) and the tissue was transversely sectioned into 300-μm-thick slices in the presence of ice-cold dissection solution. Transverse cortico-hippocampal sections were allowed to recover for ~30 min at ~31 °C and then kept at room temperature (20 ± 1°C) for the duration of the experiments. During recordings, slices were superfused with recording solution containing 125 mM NaCl, 2.5 mM KCl, 25 mM $NaHCO_3$, 1.25 mM $NaH_2PO_4$, 25 mM D-glucose, 2 mM $CaCl_2$, and 1 mM $MgCl_2$ (pH 7.4 in 95% $O_2$/5% $CO_2$, ~325 mOsm, at a rate of ~1 ml $min^{-1}$ using gravity flow). Labeled neurons in the regions of interest were identified according to their fluorescent signal and patched using pulled patch pipettes containing: 125 mM K-gluconate, 20 mM KCl, 0.1 mM EGTA, 10 mM phosphocreatine, 2 mM $MgCl_2$, 2 mM ATP, 0.4 mM GTP, 10 mM HEPES (pH adjusted to 7.28 with KOH, ~310 mOsm); 0.3% biocytin was added in a subset of recordings. Patched

cells were maintained in current-clamp mode at the neuron's resting membrane potential. Signals from patched cells were acquired using an Axon Axopatch 200 A amplifier (Molecular Devices) and digitized using a CED Power 1401 analog-to-digital converter (Cambridge Electronic Design). Optogenetic stimulation was delivered using a blue-filtered white LED (Prizmatix, IL) at maximum intensity, passed through a 63×objective which was positioned above the recorded neuron. Ten bursts, spaced 20 s apart, consisting each of 5 light pulses (5 ms in duration) at 25 Hz were delivered to the tissue. Cells which exhibited responses with latency shorter than 1 ms or longer than 10 ms were excluded from the final analysis as they likely result from direct ChIEF activation in the recorded neuron or a polysynaptic response, respectively. Following these protocols, current steps ranging from −150 pA to 300 pA, with 25 pA increments, were delivered in current-clamp mode in order to determine firing frequency and input resistance, as previously described (*Kowalski et al., 2016*). At the end of each recording session, the electrode was slowly retracted to create an outside-out patch, following which the slice was removed from the recording chamber, submerged in 4% PFA and subsequently kept in 0.1 M phosphate buffer (PB) until further processing.

## Fixed tissue preparation

For imaging purposes, animals were sacrificed 5–7 days after injection of CVS-N2c vectors. First, animals were anaesthetized as described in the previous section and intracardially perfused with 15 ml of 0.1 PB followed by 30 ml of 4% PFA. Following perfusion, the brain was removed and kept in 4% PFA overnight at 4 °C, which was subsequently replaced with 0.1 M PB. Fixed brains were sectioned 100 μm thick in either a coronal, parasagittal, or transverse plain and stored in 0.1 M PB at 4 °C.

For immunohistochemical labeling of transduced tissue, standard protocols were used. First, the sections were washed with PB 3 × /10 min. Next, sections were incubated with 10% normal goat serum (NGS) and 0.4% Triton X-100 for 1 hr, at room temperature (RT) with constant agitation and subsequently with the primary antibody (Rabbit anti-somatostatin, 1:500, BMA Biomedicals; Rabbit anti-parvalbumin, 1:1000, Swant antibodies; Chicken anti-GFP, 1:000, Abcam; Mouse anti-FLAG, 1:1000, Sigma-Aldrich), in PB containing 5% NGS and 0.4% Triton X-100, at 4 °C overnight. After washing, slices were incubated with isotype-specific secondary antibodies (Alexa Fluor 647-conjugated goat anti-rabbit, Alexa Fluor 647-conjugated goat anti-mouse or Alexa Fluor 488-conjugated goat anti-chicken) in PB containing 5% NGS and 0.4% Triton X-100 for two hours in RT with constant agitation. After washing, slices were mounted, embedded in Prolong Gold Antifade mountant (Thermo-Fisher Scientific, Cat# P36930) and sealed with a 0.5 mm coverslip.

Neurons in acute slices that were filled with biocytin (0.3%) were processed for morphological analysis. After withdrawal of the pipettes, resulting in the formation of outside-out patches at the pipette tips, slices were fixed for 12–24 hr at 4 °C in a 0.1 M PB solution containing 4% PFA. After fixation, slices were washed, treated with Streptavidin Alexa Fluor 647 Conjugate (Thermo Fisher) for two hours at room temperature, washed again and embedded in Mowiol (Sigma-Aldrich).

## Microscopy and image analysis

All representative confocal images were acquired using either an LSM 800 microscope (Zeiss) or a Dragonfly spinning disc confocal microscope (Andor). All representative confocal images displayed in this manuscript were processed using FIJI (*Schindelin et al., 2012*) and are shown as a maximal intensity projection of an image stack of 4–12 separate images. For quantification of cell numbers including channel overlap, Imaris 9 software was used.

## Cranial window implantation for in vivo CVS-N2c-GCaMP8m imaging

Before the surgery, animals were injected with meloxicam (20 mg kg$^{-1}$ s.c., 3.125 mg ml$^{-1}$ solution) and dexamethasone (0.2 mg kg$^{-1}$ i.p., 0.02 mg ml$^{-1}$ solution). Anesthesia was induced by 2.5% isoflurane in oxygen in an anesthesia chamber. The mouse was subsequently fixed in a stereotaxic device (Kopf) with constant isoflurane supply at 0.7 to 1.2% in O$_2$ and body temperature controlled by a heating pad to 37.5 °C. After assertion that reflexes subsided, the cranium was exposed and cleaned of periost and connective tissue. A circular craniotomy of 4 mm diameter was drilled above V1, careful to leave the dura mater intact and the exposed brain constantly irrigated with artificial cerebrospinal fluid. A pulled glass capillary (tip diameter 30–40 μm) was loaded with CVS-N2c-GCaMP8m (5x10$^8$ TU ml$^{-1}$) solution and 300 nl injected into the center of the craniotomy at a depth of 600 μm with a nanoliter

injector (Nanoject, World Precision Instruments) at a speed of 30 nl min⁻¹ and leaving the needle in place for 5 min after the volume was injected. Subsequently, a 4 mm circular glass coverslip (CS-4R, Warner Instruments) was positioned on the brain and careful pressure applied with a toothpick mounted in the stereotaxic arm. The glass was first fixed in place with VetBond (3 M). Then after cleaning and drying the surrounding cranium, a multilayer of glues was applied. First, to provide adhesion to the bone, All-in-One Optibond (Kerr) was applied and hardened by blue light (B.A. Optima 10). Second, Charisma Flow (Kulzer) was applied to cover the exposed bone and fix the glass in place by also applying blue light. After removal of the fixation toothpick, a custom designed and manufactured (RPD, Vienna) headplate, selective laser-sintered from the medical alloy TiAl$_6$V$_4$ (containing a small bath chamber and micro-ridges for repeatable fixation in the setup), was positioned in place and glued to the Charisma on the cranium with Paladur (Kulzer). Mice were given 300 µl of saline and 20 mg kg⁻¹ meloxicam s.c., before removing them from the stereotaxic frame and letting them wake up while kept warm on a heating pad. Another dose of 20 mg kg⁻¹ meloxicam s.c. and 0.2 mg kg⁻¹ i.p. dexamethasone was further injected 24 hr after conclusion of the surgery.

## Setup and visual stimuli for in vivo imaging

Mice were head-fixed using a custom-manufactured clamp that was connected to a 3-axis motorized stage (8MT167-25LS, Standa). Mice could run freely on a custom-designed spherical treadmill (20 cm diameter). Visual stimuli were projected by a modified LightCrafter (Texas Instruments) at 60 Hz, reflected by a quarter-sphere mirror (Modulor) below the mouse and presented on a custom-made spherical dome (80 cm diameter) with the mouse's head at its center. The green and blue LEDs in the projector were replaced by cyan (LZ1-00DB00-0100, Osram) and UV (LZ1-00UB00-01U6, Osram) LEDs respectively. A double band-pass filter (387/480 HD Dualband Filter, Semrock) was positioned in front of the projector to not contaminate the imaging. The reflected red channel of the projector was captured by a transimpedance photo-amplifier (PDA36A2, Thorlabs) and digitized for synchronization. Cyan and UV LED powers were adjusted to match the relative excitation of M- and S-cones during an overcast day, determined and calibrated using opsin templates (*Govardovskii et al., 2000*) and a spectrometer (CCS-100, Thorlabs). Stimuli were designed and presented with Psychtoolbox (*Brainard, 1997*), running on MATLAB 2020b (Mathworks). Stimulus frames were morphed on the GPU using a customized projection map and an OpenGL shader to counteract the distortions resulting from the spherical mirror and dome. The dome setup allows to present mesopic stimuli from ca. 90° on the left to ca. 170° on the right in azimuth and from ca. 40° below to ca. 80° above the equator in elevation. During anatomical stack imaging, dense moving dots of different sizes and light intensities, moving in uncorrelated directions were shown, to excite neurons with a complex texture-like stimulus. For functional imaging, full field step gratings with temporal frequency of 2 Hz and spatial frequency of 0.1 cycles/° were shown moving in 8 randomly ordered directions. In each trial, the grating image remained stationary for 3 s and then moved for 7 s in the respective direction. Each direction was shown 5–10 times in total per session.

## Imaging

Two-photon imaging was performed on a custom-built microscope, controlled by Scanimage (Vidrio Technologies) running on MATLAB 2020b (Mathworks) and a PXI system (National Instruments). The beam from a pulsed Ti:Sapphire laser (Mai-Tai DeepSee, Spectra-Physics) was scanned by a galvanometric-resonant (8 kHz) mirror combination (Cambridge Scientific) and expanded to underfill the back-aperture of the objective (16×0.8 N.A. water-immersion, Nikon); 1.9 by 1.9 mm field-of-view; 30 Hz frame rates. Fast volumetric imaging was acquired with a piezo actuator (P-725.4CA, Physik Instrumente). Emitted light was collected (FF775-Di01, Semrock), split (580 nm long-pass, FF580-FDi01, Semrock), band-pass filtered (green: FF03-525/50; red: FF01-641/75, Semrock), measured (GaAsP photomultiplier tubes, H10770B-40, Hamamatsu), amplified (TIA60, Thorlabs), and digitized (PXIe-7961R NI FlexRIO FPGA, NI 5734 16-bit, National Instruments). Laser wavelength was set to 935 or 955 nm, which excited GCaMP8 well and tdTomato sufficiently for anatomical identification. Maximum laser power used at the deepest planes was 80 mW mm⁻². Due to the early start of imaging after implantation the tissue cleared only over the course of the imaging days, necessitating relatively high laser powers in the beginning. To avoid heat damage, only 15 min of continuous imaging was performed, after which imaging was paused for at least 5 min (*Podgorski and Ranganathan, 2016*).

At each recording day (day 3, 5, 7, 9, 11, 14, 16 post RVdG$_{envA-}$CVS-N2c-GCaMP8m injection), first a dense anatomical stack with a 10 µm plane distance over the full accessible depth and plane averaging over 25 frames was recorded in the injected area. If GCaMP labeled neurons were found, subsequently a functional imaging session was started over 5–8 z-planes with 25–50 µm plane distance and voxel size of 1.4–1.7 µm resulting in a volume rate of 4.2–5 Hz.

## Imaging data analysis

Cell numbers were estimated by Imaris (Oxford Instruments) on anatomical stack images. Functional calcium imaging data was first analyzed with suite2p (v0.10.0) (*Pachitariu et al., 2016*) for motion correction and ROI extraction. ROIs were then curated manually based on morphological and activity shape. Further analysis was performed in custom MATLAB R2021a (Mathworks) scripts made available on GitLab (*Sumser, 2022*; copy archived at swh:1:rev:e55a2abf39ac9fb3592767173450c5af774218f7). dF/F0 was estimated based on published procedures (*Keller et al., 2012*) by first subtracting neuropil contamination (from suite2p, fluorescence signal of 350 pixels surrounding the ROI, excluding other ROIs) with a factor of 0.5 (estimated from fluorescence of small capillaries as reported previously); (*Kerlin et al., 2010*). From the neuropil-corrected ROI fluorescence, baseline F0 was defined as the 8th percentile of a moving window of 15 s (*Dombeck et al., 2013*). dF/F0 was then calculated on the same window by first subtracting and then dividing fluorescence trace by median of the same 15-s window (*Keller et al., 2012*). Signal-to-noise ratio (SNR) was defined for each neuron by dividing the 99th percentile of the dF/F trace ('signal') by the standard deviation of its negative values after baseline correction ('noise'). Direction selectivity index (DSI) and preferred direction was calculated based on the vector sum method (*Mazurek et al., 2014*) on the mean dF/F0 of the 7 s per direction the grating was moving. DSI significance was estimated by a permutation test of the direction labels (resampled 1000 times) to define the proportion of DSIshuffled >DSI.

## Statistical analysis

All values were reported as mean and error bars as ± SEM. Statistical significance was tested using non-parametric, single-sided Kruskal-Wallis test followed by a double-sided Mann-Whitney test for post-hoc comparisons, or by Fisher's exact test, in Microsoft Excel. Differences with $p<0.05$ were considered significant. In figures, a single asterisk (∗), double asterisks (∗∗), and triple asterisks (∗∗∗) indicate $p<0.05$, $p<0.01$ and $p<0.001$, respectively. Each experiment shown in this MS for which statistical information is not provided has been replicated in the lab a minimum of three times with identical results.

## Acknowledgements

We thank F Marr for technical assistance, A Murray for RVdG-CVS-N2c viruses and Neuro2A packaging cell-lines and J Watson for reading the manuscript. This research was supported by the Scientific Service Units (SSU) of IST-Austria through resources provided by the Imaging and Optics Facility (IOF) and the Preclinical Facility (PCF). This project was funded by the European Research Council (ERC) under the European Union's Horizon 2020 research and innovation programme (ERC advanced grant No 692692, PJ, ERC starting grant No 756502, MJ), the Fond zur Förderung der Wissenschaftlichen Forschung (Z 312-B27, Wittgenstein award, PJ), the Human Frontier Science Program (LT000256/2018-L, AS) and EMBO (ALTF 1098-2017, AS).

## Additional information

### Funding

| Funder | Grant reference number | Author |
|--------|------------------------|--------|
| Horizon 2020 Framework Programme | 692692 | Peter Jonas |
| Austrian Science Fund | Z 312-B27 | Peter Jonas |

| Funder | Grant reference number | Author |
| --- | --- | --- |
| Human Frontier Science Program | LT 000256-2018-L | Anton Sumser |
| Horizon 2020 Framework Programme | 756502 | Maximilian Joesch |
| EMBO | ALTF 1098-2017 | Anton Sumser |

The funders had no role in study design, data collection and interpretation, or the decision to submit the work for publication.

### Author contributions
Anton Sumser, Formal analysis, Methodology; Maximilian Joesch, Peter Jonas, Supervision; Yoav Ben-Simon, Conceptualization, Data curation, Formal analysis, Validation, Methodology, Writing – original draft, Investigation, Project administration, Visualization, Writing – review and editing

### Author ORCIDs
Anton Sumser ⓘ http://orcid.org/0000-0002-4792-1881
Maximilian Joesch ⓘ http://orcid.org/0000-0002-3937-1330
Peter Jonas ⓘ http://orcid.org/0000-0001-5001-4804
Yoav Ben-Simon ⓘ http://orcid.org/0000-0002-7075-097X

### Ethics
Experiments on C57BL/6 wild-type and transgenic mice were performed in strict accordance with institutional, national, and European guidelines for animal experimentation and were approved by the Bundesministerium für Wissenschaft, Forschung und Wirtschaft and Bildung, Wissenschaft und Forschung, respectively, of Austria (A. Haslinger, Vienna; BMWF-66.018/0010-WF/V/3b/2015; BMBWF-66.018/0008-WF/V/3b/2018; BMWFW-66.018/0017-WF/V/3b/2017).

### Decision letter and Author response
Decision letter https://doi.org/10.7554/eLife.79848.sa1
Author response https://doi.org/10.7554/eLife.79848.sa2

## Additional files

### Supplementary files
• MDAR checklist

### Data availability
All source data associated with the manuscript has been included in the manuscript.

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
