## [Editor Report]

Rabies-mediated monosynaptic retrograde tracing is a powerful method to characterize the connectivity of neural circuits. The CVS-N2c strain of rabies virus shows significantly higher efficiency of transsynaptic spread and less toxicity than the more commonly used SAD B19 strain but has been limited in use by an arduous and lengthy packaging process and low resultant titers. Here, Sumser et al. present a method that significantly speeds up the production process while reducing off-target expression. They also introduce a suite of novel reagents (34 novel plasmids) for monosynaptic tracing with the CVS-N2c strain that they commendably, have already deposited with Addgene. The work is an important advance that will reinvigorate rabies-mediated circuit tracing.

---

## [Decision Letter]

**Decision letter after peer review:**

Thank you for submitting your article "Fast, high-throughput production of improved rabies viral vectors for specific, efficient and versatile transsynaptic retrograde labeling" for consideration by *eLife*. Your article has been reviewed by 3 peer reviewers, and the evaluation has been overseen by Rebecca Seal as the Reviewing Editor and Laura Colgin as the Senior Editor. The following individual involved in the review of your submission has agreed to reveal their identity: Ian Wickersham (Reviewer #2).

Essential revisions:

1) The authors claim that the titer efficiency is high but do not clearly show the titers which should be calculated for virus harvested from cells expressing TVA and from cells not expressing TVA.

2) The authors claim that there is no contamination with G-coated rabies but do not show it. The authors need to demonstrate the level of background contamination by injecting the rabies without the TVA in WT mice. Leak expression from Cre-dependent "starter" AAVs should also be tested by injecting them and rabies in WT mice.

3) To improve the clarity of the manuscript, the authors should address all other points outlined by the reviewers below, which will not require any new experiments.

*Reviewer #1 (Recommendations for the authors):*

The authors should examine the time course of health of the retrogradely infected cells and starter cells with electrophysiological measurements of membrane potential or a histological stain.

Extended Data Figure 7 seems errantly labeled in C. Number of neurons?

Abbreviations in figures should be defined in figure legends.

---

## [Author Response]

Reviewer #1 (Recommendations for the authors):The authors should examine the time course of health of the retrogradely infected cells and starter cells with electrophysiological measurements of membrane potential or a histological stain.

We thank the reviewer for this suggestion. We have added resting potential information of retrogradely infected cells and starter cells at ~ 7 days post injection (Figure 6D).

Regarding the time course of neuronal health beyond this point, the situation is more complex. Reardon et al., 2016 reported anecdotal evidence that channelrhodopsin stimulation was possible until 28 days after infection. However, other assays of integrity in this paper showed signs of degeneration at much earlier stages (e.g. increase of propidium iodide-positive cells 4 days after infection). Although we have obtained patch recordings at later times after injection, we had the impression that the – number – of labeled cells decreased, and that it is more difficult to get recordings. However, this is difficult to quantify, because of the high variability of labelling across animals. Furthermore, it is well known that neurons become less healthy and more difficult to record by electrophysiology in older animals, even in wild-type. Thus, after careful consideration, we arrived at the conclusion that obtaining compelling data will be difficult.

We also want to emphasize that the rabies virus-mediated expression of both ChIEF and GCaMP8 is highly powerful and has a fast onset. Therefore, we are sure that both actuator and/or indicator expression reach sufficiently high levels after ~5 days. Thus, although the health of neurons may decline at later times after infection, there is a large window in which meaningful functional connectivity measurements can be obtained. We have now mentioned this in the Results / Discussion section of the revised version (p. 11, top of the revised manuscript).

We hope that the reviewer can accept our reply, which represents an honest answer to a tricky question.

Extended Data Figure 7 seems errantly labeled in C. Number of neurons?

ED Figure 7c (now Figure 7 —figure supplement 9C) indeed shows the development of fluorescence SNR over time. To avoid confusion, the text describing the results has been revised to enable easier interpretation.

Abbreviations in figures should be defined in figure legends.

We have carefully reviewed all figure legends and made sure that all abbreviations are properly defined.